# Speculative Coupled Decoding for Training-Free Lossless Acceleration of Autoregressive Visual Generation

Junhyuk So [1]   Hyunho Kook [1]   Chaeyeon Jang [1]   Eunhyeok Park [1 2]

## Abstract

Autoregressive (AR) modeling has recently emerged as a promising new paradigm in visual generation, but its practical adoption is severely constrained by the slow inference speed of per-token generation, which often requires thousands of steps to produce a single sample. While several Speculative Decoding (SD)-based methods have been proposed to solve this problem by generating multiple tokens in a single forward step, they suffer from limited speedup, degraded quality, or require the training of a draft model. To solve these problems, we propose a new training-free, lossless SD framework, Speculative Coupled Decoding (SCD), by extending the recently proposed Speculative Jacobi Decoding (SJD). While SJD shows strong potential for accelerating AR generation by combining Jacobi iteration and SD, we found that its acceptance rate is still significantly limited due to the instability arising from the independent sampling process used during draft token generation. To overcome this, we introduce an information-theoretic approach, Coupling, which stabilizes the drafting trajectory of SJD by maximizing the probability of sampling identical draft tokens across consecutive iterations, significantly enhancing the acceptance rate while preserving its lossless property. Remarkably, this method requires only a single-line modification to the existing algorithm with almost zero overhead, yet achieves substantial performance gains, delivering up to a 4.2× speedup in image generation and 13.6× speedup in video generation compared to standard AR decoding, without any degradation or the need for additional training. The source code is available at https://github.com/junhyukso/SCD.

[1]Department of Computer Science and Engineering, POSTECH, South Korea [2]Graduate School of Artificial Intelligence, POSTECH, South Korea. Correspondence to: Eunhyeok Park <eh.park@posech.ac.kr>.

*Proceedings of the 43rd International Conference on Machine Learning*, Seoul, South Korea. PMLR 306, 2026. Copyright 2026 by the author(s).

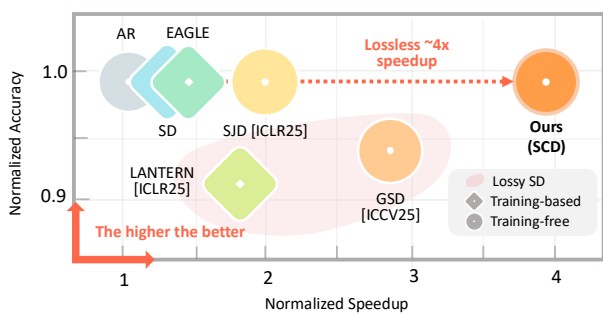

*Figure 1.* Comparison of recent SD methods for AR image generation. While recent works suffer from limited acceleration or sacrifice quality, our **SCD** achieves up to 4.2x speedup over standard AR decoding without any quality degradation.

## 1. Introduction

Recently, autoregressive (AR) modeling has emerged as a cornerstone of modern generative AI (Brown et al., 2020; Achiam et al., 2023), achieving state-of-the-art performance not only in text generation (Touvron et al., 2023) but also across diverse modalities including images (Liu et al., 2024; Sun et al., 2024a), video (Agarwal et al., 2025), 3D meshes (Weng et al., 2025), audio (Du et al., 2024; Wang et al., 2023), and even robotics (Pertsch et al., 2025). Its key strength lies in the ability to unify training and inference across modalities within a single framework, enabling flexible generation, editing, and translation. This cross-domain unification allows models to leverage rich knowledge from different sources, enhancing both understanding and generation (Zhang et al., 2025).

However, the practical power of AR modeling is often constrained by the inherent cost of massive computation and exacerbated memory bottlenecks. Generating a sequence of $N$ tokens requires $N$ AR forward passes, leading to significant latency. The problem becomes particularly severe for high-dimensional data such as images and video, where thousands of tokens are needed to represent a single high-resolution instance, acting as a critical barrier to the real-world deployment of multimodal AR models at scale.

Recently, speculative decoding (SD) (Leviathan et al., 2023) has been actively explored (Sun et al., 2023; Yin et al., 2024) to solve this problem, particularly for large language

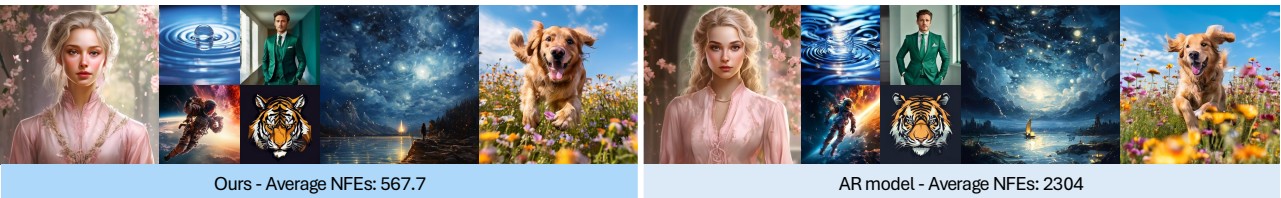

Ours - Average NFEs: 567.7      AR model - Average NFEs: 2304

*Figure 2.* Qualitative comparison between **Ours (SCD)** v.s. *Vanilla AR* on Lumina-mGPT. (zoom-in to view).

models (LLMs) in text generation. The core idea is to use computationally smaller draft model to propose multiple candidate tokens fast, which are then verified in parallel by the accurate target model. More importantly, SD is a ***lossless*** acceleration method, guaranteeing that its output distribution remains theoretically identical to standard AR decoding. However, despite its effectiveness, SD has notable drawbacks: the overhead of training a separate draft model (Cai et al., 2024), and limited performance in vision generation tasks (So et al., 2025; Jang et al., 2024).

To address these problems, the Speculative Jacobi Decoding (SJD) (Teng et al., 2024) was proposed, combining Jacobi iteration (Song et al., 2021) with the stochastic verification criterion of SD. SJD uses the output distribution from its own previous verification step as the draft for the next, eliminating the need for a separate, trained draft model, thereby resolving the idle-time bottleneck and demonstrating significant speedups, especially in image generation. However, while SJD shows promise, it delivers only about $\sim$2$\times$ speedup in image generation, relatively modest compared to state-of-the-art SD methods in text generation, which achieve over 4$\times$ acceleration (Cai et al., 2024).

In this paper, we demonstrate that this problem can be solved with a simple tweak to the SJD process, offering an incredibly high speedup while maintaining the lossless property of SD. Our key finding is that the performance of the SJD is significantly limited by instability in its draft token sampling, leaving substantial room for further improvement. To unlock the potential of SJD, we introduce a simple yet highly effective, information-theory-inspired idea: **Coupling** (Lindvall, 2002). Specifically, we propose to couple the draft sampling process between consecutive Jacobi iterations, thereby increasing the probability of sampling identical tokens to promote stability. This method requires only a single-line modification to the standard SJD, making it extremely simple to implement without any additional training. Despite its simplicity, we demonstrate that our method significantly enhances the acceptance rate of SJD, enabling a remarkable speedup of $\sim$3.8$\times$ for AR image generation and $\sim$10$\times$ for video generation, compared to standard AR decoding.

## 2. Preliminaries

**Notation.** We denote by $X_i^t$ the token at the $i$-th position of a sequence $X$ at Jacobi iteration $t$ (defined later). When

---

**Algorithm 1** MRS(p,q,x); Modified Rejection Sampling

**Input**: Distribution $P$, $Q$. Tokens $X \sim Q$
**Output**: Random variable $Y$, Accept signal $k$.
1: Sample $u \sim \mathcal{U}[0,1]$
2: **if** $u \le \min(1, \frac{P(X)}{Q(X)})$
3:     **return** Y=X, 1
4: **return** $Y \sim norm(max(0, P(x) - Q(x))), 0$

---

clear from context, we omit the subscript/superscript $i$ or $t$ to refer to the entire sequence or to the collection of distributions, respectively. Similarly, we denote by $p_i^t(\cdot)$ the token distribution at position $i$ in iteration $t$. We assume all distributions are on the same support $\mathcal{V}$.

### 2.1. Speculative Decoding

Speculative Decoding(SD) (Leviathan et al., 2023) is primary technique for accelerating LLMs in the text-generation domain. The main goal of SD is to reduce the number of sequential calls of target model $p$ while ensuring that final outcome matches the original token-by-token Autoregressive (AR) sampling distribution, $\prod_i p_i(x \mid X_{<i})$. Specifically, we assume two models: a target $p(x)$, which we wish to accelerate, and a draft $q(x)$, which is faster than $p(x)$ but less accurate. SD proceeds as follows:

1. **Drafting**: Sample $L$ draft tokens from the draft distributions, $X_{i:i+L-1} \sim q_{i:i+L-1}(\cdot)$.

2. **Evaluate**: Target model evaluate token probabilities along the drafted prefixes $\{ p_j(X_j \mid X_{<j}) \}_{j=i}^{i+L}$.

3. **Verify**: Run Algorithm 1 with $(p_i, q_i, X_i)$ sequentially until a rejection occurs (i.e., the procedure returns $k = 0$); Then *accept* all previously verified tokens.

4. **Repeat**: If the generation is not yet complete, return to **Drafting** and repeat the process.

Transformers natively support the *parallel* evaluation in step (2) via masked attention, ideally in $O(1)$ sequential depth. Thus, if *acceptance* occurs in step (3), this procedure emit multiple tokens in effectively $O(1)$ sequential time, reducing total NFEs compared with standard AR decoding.

Notable advantage of SD is ***lossless*** acceleration. The sampling of Alg. 1 guarantees that even if the input is $X \sim q(\cdot)$, the output $Y$ returned by the algorithm satisfies $Y \sim p(\cdot)$

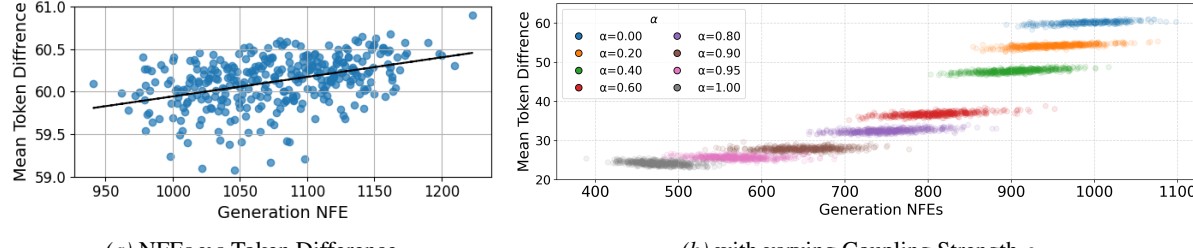

*(a)* NFEs v.s Token Difference        *(b)* with varying Coupling Strength $\alpha$

*Figure 3.* (a) Generation NFE v.s. Mean Token Difference during SJD with window size $L = 64$. As shown, a sample that generated with small NFE tends to have small mean token difference. (b) Increasing the coupling strength $\alpha$ decreases Mean Token Difference and NFEs until $\alpha \approx 1$, indicating that pushing collision probability to its maximal tends to have significant positive effect on minimizing total NFEs.

(Chen et al., 2023). Because each Markov chain follows valid sampling from $p$ until the first rejection occurs, the theoretical correctness of speculative decoding is ensured. As shown in Alg. 1, the acceptance probability per token, $\min\{1, p(x)/q(x)\}$, is the key factor that determines the overall speedup. We formalize it in following proposition:

**Proposition 2.1.** *Let $q$ be the draft distribution and $x \sim q(x)$, then, final output from MRS(Alg. 1) strictly follow the distribution of target model $p(x)$. Moreover, the acceptance rate of this algorithm is defined as*

$$\mathbb{E}_{x \sim q(x_i)} min(1, \frac{p(x)}{q(x)}) = 1 - \mathcal{D}_{TV}(p, q)$$

*where $\mathcal{D}_{TV}$ denotes total variation $\frac{1}{2} \sum_v |p(v) - q(v)|$.*

*Proof:* See appendix. Typically, standard SD methods employ a "cheaper" AR model to produce draft tokens/distributions. While this strategy has shown promising results in the text-generation domain, it has several drawbacks: the need to train a separate draft model, communication bottlenecks between the draft and target models, and limited speedups in non-text AR generation domains (So et al., 2025; Jang et al., 2024). These issues have hindered the adoption of SD techniques beyond text, limiting the potential of AR modeling across different modalities.

### 2.2. Speculative Jacobi Decoding

Speculative Jacobi Decoding (SJD) (Teng et al., 2024) is a recently proposed training-free SD to solve the aforementioned problems of SD. As depicted in Alg. 2, SJD eliminates the need for a separate draft model $q(\cdot)$ by leveraging the probability distribution from its own previous validation step as the draft for the next iteration. This ***Self-SD*** approach does not impact the lossless property of SD, because the verification mechanism (Alg. 1) ensures the output is always a valid sample from the target model's distribution, regardless of the input draft. This framework makes the process highly efficient as it removes the overhead of training a separate model and eliminates the idle time where the target model would wait for draft tokens. Due to these properties, SJD first achieves a $\sim$2x speedup in AR image generation

domain, while retaining its lossless and training-free nature.

## 3. Motivation and Analysis

Despite SJD achieves meaningful speedup in image AR, we find that its performance potential is significantly limited by the variance introduced during its stochastic draft sampling process. To gain an intuitive understanding of this, we start with a formal analysis of the acceptance rate of SJD. At iteration $t$ in SJD, the target distribution is $p^t(\cdot)$ and the draft distribution is $p^{t-1}(\cdot)$. As noted in Proposition 2.1, the acceptance rate can be expressed in terms of the Total Variation, as follows :

$$\beta_i^{(t)} = 1 - \mathcal{D}_{TV}\left(p_i^{(t)}(x), p_i^{(t-1)}(x)\right) \quad (1)$$

$$= 1 - \mathcal{D}_{TV}\left(p_\theta\left(\cdot \mid X_{<i}^{(t-1)}\right), p_\theta\left(\cdot \mid X_{<i}^{(t-2)}\right)\right), \quad (2)$$

where $p_\theta$ denotes the autoregressive model and $X_{<i}$ denotes the prefixes $\{X_{i-1}, X_{i-2}, \dots\}$. As shown in Eq. 2, The acceptance rate $\beta_i^{(t)}$ is directly influenced by the *context change* between iterations $t - 1$ and $t - 2$. In other words, the acceptance rate for token $i$ is driven by changes in its *prefixes*, including both previously accepted tokens but also the other rejected tokens in the draft. This leads directly to the following observation:

**Observation 3.1.** *High context similarity between consecutive drafts tends to yield a higher speedup.*

This can be easily validated by Eq. 2: the greater the similarity between the contexts $X_{<i}^{(t-1)}$ and $X_{<i}^{(t-2)}$, the more similar their corresponding output distributions $p_\theta(\cdot|X_{<i})$ will be, under a mild Lipschitzness assumption. This results in a lower TV distance and, consequently, a higher acceptance rate $\beta$. We also empirically validate it in Fig. 3a, plotting the 300 independent samples with their mean number of changed tokens between consecutive sequence drafts (Hamming distance) against the total number of function evaluations (NFE) required for SJD generation. As shown, there is a strong correlation between these two, indicating that context similarity plays a crucial role for faster generation in SJD.

---

**Algorithm 2** Speculative Jacobi Decoding

---

**Require:** AR Model $p_\theta$, Window len. $L$, Sequence Len $N$
1: $p_t^i \leftarrow$ Random()
2: $X_i^t \sim p_i^t \quad \forall i, t$ $\quad\quad\quad\quad\quad\quad\quad\quad$ ◁ Initialize

3: **While** $i < N$
4: $\quad$ **parallel for** $j = i$ to $i + L$ : $\quad\quad\quad$ ◁ Drafting
5: $\quad\quad X_j^t \sim p_j^t(x)$

6:

7: $\quad$ **parallel for** $j = i$ to $i + L$ : $\quad\quad\quad$ ◁ Evaluate
8: $\quad\quad p_j^{t+1} \leftarrow p_\theta(\cdot \mid X_{0:j-1}^t)$
9: $\quad$ **for** $j = i$ to $i + L$ : $\quad\quad\quad\quad\quad$ ◁ Verify
10: $\quad\quad k, X_j^{t+1} \leftarrow \text{MRS}(p_j^{t+1}, p_j^t, X_j^t)$
11: $\quad\quad$ **if** $k = 0$ : **break;**
12: $\quad i \leftarrow j + 1, t \leftarrow t + 1$
13: **Return** $X$

---

**Algorithm 3** Pseudo Code for our **SCD**

---

**Require:** AR Model $p_\theta$, Window len. $L$, Sequence Len $N$
1: $p_t^i \leftarrow$ Random()
2: $X_i^t \sim p_i^t \quad \forall i, t$ $\quad\quad\quad\quad\quad\quad\quad\quad$ ◁ Initialize

3: **While** $i < N$
4: $\quad$ **parallel for** $j = i$ to $i + L$ : $\quad\quad\quad$ ◁ Drafting
5: $\quad\quad X_{j,\_}^t \leftarrow \text{MRS}(p_j^t, p_j^{t-1}, X_j^t)$ $\quad$ **Maximal Coupling**

6: $\quad\quad X_{j,\_}^t \leftarrow \text{GS}(p_j^t, p_j^{t-1}, G_j)$ $\quad\quad$ **Gumbel Coupling**

7: $\quad$ **parallel for** $j = i$ to $i + L$ : $\quad\quad\quad$ ◁ Evaluate
8: $\quad\quad p_j^{t+1} \leftarrow p_\theta(\cdot \mid X_{0:j-1}^t)$
9: $\quad$ **for** $j = i$ to $i + L$ : $\quad\quad\quad\quad\quad$ ◁ Verify
10: $\quad\quad k, X_j^{t+1} \leftarrow \text{MRS}(p_j^{t+1}, p_j^t, X_j^t)$
11: $\quad\quad$ **if** $k = 0$ : **break;**
12: $\quad i \leftarrow j + 1, t \leftarrow t + 1$
13: **Return** $X$

---

However, despite this correlation, Fig. 3a shows that the average number of token difference is approximately 94% (60 of window size 64), indicating significantly large portion of tokens are changed in each iteration. We observe that this high degree of change not only critically limits SJD's acceptance rate but also poses a more severe problem when considering the behavior over multiple consecutive iterations, the *convergence* of SJD.

**Observation 3.2.** *The per-token acceptance rate $\beta_i^t$ during the SJD process exhibits high variance and does not show converging behavior.*

Ideally, the acceptance rate for a given token should increase over the SJD iterations. As the left-most context becomes filled with stable, accepted tokens, an *improvement signal* should propagate to the right, progressively enhancing the quality of the draft sequences. However, our empirical results reveal the opposite behavior. Fig. 4a plots the trajectory of $\beta_i^t$ for representative tokens, showing that the acceptance rate frequently fluctuates without any consistent upward trend. This instability is further confirmed in Fig. 4c (blue line), which aggregates the statistics across all tokens. After an initial jump, the mean acceptance rate not only remains low but also fails to improve, exhibiting random fluctuations with high variance throughout the process.

### 3.1. Analysis

We then investigate the root cause of this low context similarity. Since the context sequences $X^{(t)}$ are realizations of random variables drawn from $p^{(t)}(\cdot)$ at each iteration, a natural way to quantify their similarity is by measuring the **collision probability**, defined as $\Pr[X_i^{(t)} = X_i^{(t-1)}]$. As described in Alg. 2, the drafting stage of SJD (Line' 5) samples the draft token $X_i^{(t)}$ *independently* from its distribu-

tion $p_i^t(x)$. In this scheme, the collision probability between $X_i^{(t)}$ and $X_i^{(t-1)}$ can be analytically computed as follows:

**Proposition 3.3** (SJD Collision Probability)**.** *Standard SJD has following collision probability for token $i$ at iteration $t$:*

$$C_{SJD}(p^{(t)}, p^{(t-1)}) = \sum_{x \in \mathcal{V}} p_i^{(t)}(x) \cdot p_i^{(t-1)}(x)$$

*where $\mathcal{V}$ denotes the vocabulary. This value is bounded as :*

$$C_{SJD}(p, q) \leq e^{-1/2 \cdot (H_2(p) + H_2(q))}$$

*where $H_2(p) = -log(\sum_x p(x)^2)$ is Rényi-2 entropy of $p$.*

*Proof:* See appendix. As shown, even when two distributions are similar, their collision probability is constrained by the (Rényi-2) entropy of the underlying distributions. Unfortunately, unlike text AR models, visual AR models are known to generate very flat distributions (So et al., 2025). This is because of the inherent redundancy in visual tokens and the complexity of visual patterns makes a large number of different tokens as a plausible continuations of sequence.

We also visualize the empirical collision probability, $C_{SJD}$, during the SJD process. Fig.5a shows most of its values remain at an extremely low value and Fig.5c illustrates that these are nearly zero regardless of whether $\mathcal{D}_{TV}$ is small. Consequently, standard SJD propagates different contextual information to subsequent tokens in each iteration, significantly destabilizing the convergence and causing to fluctuate unpredictably. We identify this discrepancy between the proximity in probability space and the realized token space as the *key factor* limiting the speedup in SJD.

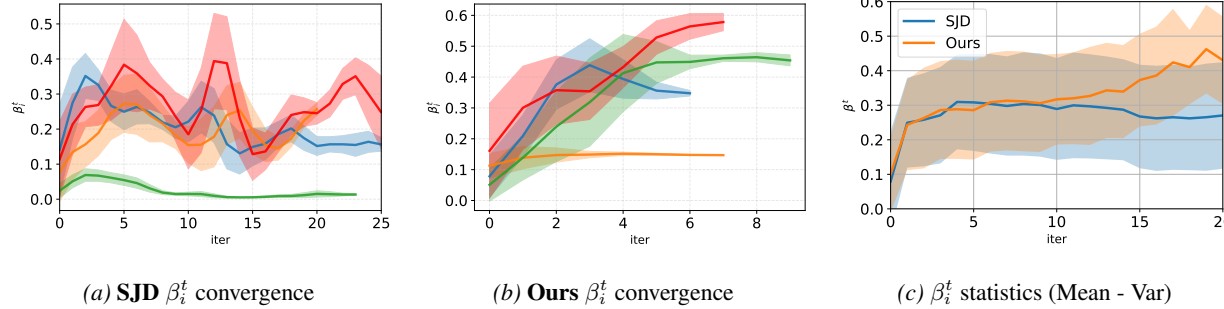

*(a)* **SJD** $\beta_i^t$ convergence  *(b)* **Ours** $\beta_i^t$ convergence  *(c)* $\beta_i^t$ statistics (Mean - Var)

*Figure 4.* (a), (b) The trajectory of tokenwise acceptance rate $\beta_i^t$ during the jacobi iterations  (a) Standard SJD shows most tokens have large variation during iteration and do not exhibit improvement behavior. (b) After applying our coupled sampler $\pi_{MC}$. Now most of tokens has very small fluctuation, showing general upward trends. (c) Mean and variance of $\beta_i^t$ across all token index. While standard SJD does not show improvement, ours shows clear upward, refining behavior.

## 4. Methods

Our main idea is that making **Coupling** (Lindvall, 2002) between the draft distributions from consecutive iterations can increase collision without compromising the theoretical lossless property of the SD. To formalize, we begin with mathematical definition of coupling :

**Definition 4.1** (Coupling). *For two distributions $P(\cdot)$ and $Q(\cdot)$ on the same support $\mathcal{V}$, a joint distribution $\pi(\cdot, \cdot)$ over $\mathcal{V} \times \mathcal{V}$ is a **Coupling** of $P$ and $Q$ if its marginals satisfy:*

$$\sum_{y \in \mathcal{V}} \pi(x, y) = P(x) \quad and \quad \sum_{x \in \mathcal{V}} \pi(x, y) = Q(y)$$

Our key insight lies in the *marginalization* property of a coupling. If we sample a pair of variables from a joint distribution $\pi(x, y)$, the marginal distribution of each individual variable remains identical to its original distribution (e.g., $P(x)$ and $Q(y)$). Therefore, using a token sampled from a coupling is a provably valid replacement for independent sampling within the SJD framework. We formally stated it in the following theorem:

**Theorem 4.2.** *Let $\Pi_i^{(t)}$ be the set of all possible couplings between $p_i^{(t)}$ and $p_i^{(t-1)}$. If we sample a pair $(X_i^{(t)}, X_i^{(t-1)}) \sim \pi(\cdot, \cdot)$ for any $\pi \in \Pi_i^{(t)}$ and use $X_i^{(t)}$ as the draft token in Algorithm 1, the final output distribution still correctly matches the target model's distribution.*

*Proof Sketch: See appendix.* For any given coupling $\pi$, we can define its effectiveness using a metric called as **Coupling Cost**, denoted $C(\pi)$. This cost measures the probability of sampling identical variables from the joint distribution, which is actually the same metric we previously referred to as the collision probability:

**Definition 4.3** (Coupling Cost). *Let $\pi_{P,Q}$ be a Coupling of distributions $P$ and $Q$ as per Definition 4.1. The Coupling Cost $C(\pi_{P,Q})$ is defined as:*

$$C(\pi_{P,Q}) = \Pr_{(X,Y) \sim \pi_{P,Q}} [X = Y] = \mathbb{E}_{(X,Y) \sim \pi_{P,Q}} \mathbb{1}\{X = Y\}$$

From this perspective, the standard SJD process can be understood as using an *independence coupling*, where $\pi_{SJD}(x, y) = p_i^{(t)}(x) \cdot p_i^{(t-1)}(y)$ and the cost of this coupling is $C(\pi_{SJD}) = \sum_v p_i^{(t)}(v) p_i^{(t-1)}(v)$, a value we have already shown to be extremely low in AR image generation. Finally, our main objective can be safely reframed as finding an alternative coupling, $\pi^*$, that **maximizes** this cost, thereby promoting context similarity without compromising the exactness guarantee of the framework. We next present alternative couplings that achieve this objective.

### 4.1. Maximal Coupling

Consider the computation graph of $X^t$ and $p^t(x)$ during the SJD process :

$$X^{t-2} \to p^{t-1}(X) \to X^{t-1} \to p^t(X) \to X^t \quad (3)$$

As shown, at the time of sampling step for $X^{(t)}$, we already have access to the full information of two distributions $p^{(t)}$ and $p^{(t-1)}$. As is well-established in many literature on information theory and optimal transport (Villani et al., 2008; Bavarian et al., 2016), having complete information of both distributions allows us for the construction of a **maximal coupling**, which has the cost of $c(\pi_{p,q}) = 1 - \mathcal{D}_{TV}(p, q)$ that any two distribution can maximally have.

In Alg. 3, we present the implementation of our SCD. As shown, the only modification required is in the drafting phase (Line' 5), where we now sample the draft token $X^t$ using a *coupled sampler* instead of sampling it independently from $p_i^t(\cdot)$. Interestingly, as shown, the implementation of maximal coupling is exactly identical with the modified rejection sampling (MRS$(\cdot)$, Alg.1) which we used for SD verification process. This can be easily validated by the fact that MRS$(\cdot)$ returns $Y \sim P$ from an input $X \sim Q$, ensuring that the marginals of the generated pair $(Y, X)$ match $P$ and $Q$, which satisfies the definition of a coupling. Moreover, as established in Proposition 2.1, the acceptance rate MRS$(\cdot)$ - probability of $\Pr[X = Y]$ - is $1 - \mathcal{D}_{TV}(p, q)$. This value is the theoretical upper bound of coupling cost,

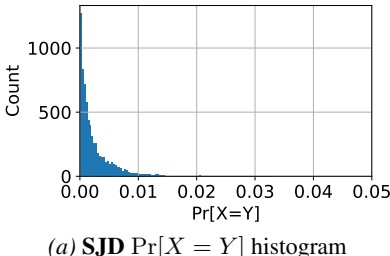

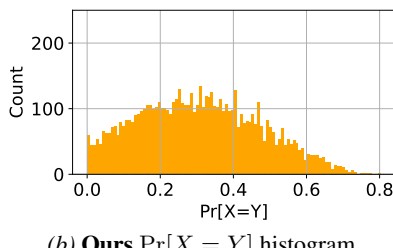

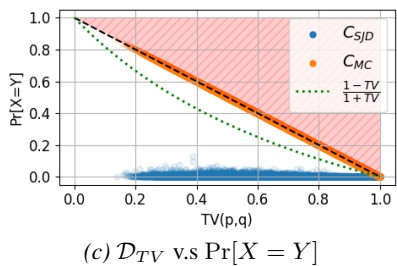

| *(a)* **SJD** $\Pr[X=Y]$ histogram | *(b)* **Ours** $\Pr[X=Y]$ histogram | *(c)* $\mathcal{D}_{TV}$ v.s $\Pr[X=Y]$ |

*Figure 5.* Visualization of Collision probabilities. (a) During standard SJD, $C_{SJD}$ are concentrated on extremely small values. (b) Our Coupler elevates this to much higher values, significantly enhancing the context similarity. (c) Standard SJD has a low $\Pr[X=Y]$ even when the corresponding TV distance is low. The green dot-line denotes the $\pi_{GS}$ lower bound $\pi_{GS} \geq (1 - \mathcal{D}_{TV})/(1 + \mathcal{D}_{TV})$.

confirming that this procedure is maximal coupling. We formally state this as follows:

**Theorem 4.4.** *Let the pair (X,Y) be generated by Algorithm 1. Then, their resulting joint distribution* $(X, Y) \sim \pi_{MC}$*, is a valid coupling of P and Q. Moreover, its coupling cost,*

$$C(\pi_{MC}) = 1 - \mathcal{D}_{TV}(P,Q)$$

*is the upper bound for the cost of any* $\pi \in \Pi$ *with P and Q.*

As illustrated in Fig. 5c, this upper bound, represented by the black dashed line, shows a significant gap compared to the coupling cost of standard SJD ($C_{SJD}$). Applying maximal coupling within SJD, elevates this low values to their upper bound (orange dots), thereby strongly promoting high context similarity and achieving a greater speedup. We also show the distribution of $C(\pi_{MC})$ in Fig. 5b. In Fig. 4b,4c, we show the trajectories and statistics of the $\beta^t$, during iterations with our SCD($\pi_{MC}$). As shown, most tokens now exhibit minimal fluctuation with a general upward trend, resulting in a much higher overall acceptance rate compared to standard SJD, leading to lower NFEs.

---

**Algorithm 4** GS(P, Q, G); Gumbel Sharing Coupling

**Input**: Distributions $P, Q$ over a vocabulary $\mathcal{V}$. Gumbel noise vector $G = (g_1, \ldots, g_{|\mathcal{V}|})$ where $g_i \sim \text{Gumbel}(0,1)$.
**Output**: Coupled Random Variables $(X, Y)$.

  1: $X \leftarrow \text{argmax}_{i \in \mathcal{V}}(\log(P_i) + g_i)$
  2: $Y \leftarrow \text{argmax}_{i \in \mathcal{V}}(\log(Q_i) + g_i)$
  3: **return** $(X, Y)$

---

### 4.2. Gumbel Coupling

In Alg. 4, we also propose *Gumbel Coupling* ($\pi_{GS}$), which have similar coupling cost with $\pi_{MC}$ but have different characteristic and convergence behavior. As shown, this algorithm is based on the Gumbel-Max trick that relies on sharing the same random noise vector to couple two categorical sampling processes. GS versioned SCD can also be seen in Alg. 3. We establish its validity and provide a lower bound for its cost:

**Theorem 4.5.** *Let the pair (X,Y) be generated by Algorithm 4. Then, their resulting joint distribution* $(X, Y) \sim$

$\pi_{GS}$*, is a valid coupling of P and Q. Its worst-case coupling cost is lower-bounded by:*

$$C(\pi_{GS}) \geq (1 - \mathcal{D}_{TV}(P,Q))/(1 + \mathcal{D}_{TV}(P,Q))$$

*Proof : See appendix.* As shown in Fig. 5c, this lower bound is significantly greater than the independence coupling $\pi_{SJD}$ and almost tight to the maximal $1 - \mathcal{D}_{TV}$ line, thus achieving performance comparable to $\pi_{MC}$. Moreover, $\pi_{GS}$ can even yield better NFEs in some tasks. Since this lower bound is applicable to any pair of distributions during an iteration, this Gumbel Coupling promotes more *long-range stabilization*, whereas $\pi_{MC}$ can be seen as greedy optimization within each consecutive iteration. We found that this behavior can be beneficial in tasks where draft prediction is relatively easy, so that keeping early draft tokens unchanged can have more beneficial effect than continually updating them with slightly more accurate information, as discussed in more detail in the Sec. 5.1.

### 4.3. Implementation Details

One of the primary advantage of our method is that these *Coupler* introduce almost no overhead compared to standard SJD, in terms of both mem-

*Table 2.* Latency breakdown ($ms$) per NFE step (Janus-Pro 7B, RTX 3090).

| Operation | $L=16$ | $L=32$ | $L=64$ |
|---|---|---|---|
| Preprocessing | 1.52 | 1.53 | 1.58 |
| **Transformer Fwd** | **26.49** | **27.73** | **36.41** |
| Logit Proc. | 0.16 | 0.23 | 0.25 |
| **Token Sampling (GS)** | **0.13** | **0.13** | **0.14** |
| **Vec. MRS (MC)** | **1.56** | **1.58** | **1.66** |
| Post Processing | 0.81 | 0.82 | 0.84 |

ory and computation. Please see the actual efficient implementation version of our SCD in Alg. 6,7. In efficient version, $\pi_{MC}$ can be implemented by simply vectorizing verification loop(Alg. 3, Line' 10) without *break* and saving the index where the first rejection ($k = 0$) occurs, thereby integrating *drafting and verification into a single operation* to minimize overhead. This can be easily validated by the fact that Line' 5 and Line' 10 in Alg. 3 are strictly identical operations because $p^{(t+1)}$ in Line' 10 becomes $p^{(t)}$ in next loop's Line' 5. Similarly, the Gumbel noise in $\pi_{GS}$ can be efficiently generated online via hashing of the token's global index, and its computation cost is identical to multinomial sampling in $\pi_{SJD}$. In Tab. 2, we present a

*Table 1.* Quantitative evaluation results on AR Image generation, Lumina-mGPT (Liu et al., 2024) on MS-COCO dataset.

| | Configuration | NFE ($\downarrow$) | Latency ($\downarrow$) | Acceleration ($\uparrow$) | | FID ($\downarrow$) | IS ($\uparrow$) | CLIP-Score ($\uparrow$) |
|---|---|---|---|---|---|---|---|---|
| | | | (A100) | NFE | Latency | | | |
| A | *Vanilla AR* | 2390 | 102.03s | 1.00× | 1.00× | 30.79 | 32.81 | 31.31 |
| B | SJD (L=16) | 1058.6 | 43.02s | 2.25× | 2.37x | 30.77 | 32.78 | 31.32 |
| D | + **Ours** ($\pi_{MC}$) | **814.5** | **32.75s** | **2.94×** | **3.12x** | 30.73 | 33.56 | 31.32 |
| D | + **Ours** ($\pi_{GS}$) | 819.4 | 32.89s | 2.92× | 3.10x | 30.78 | 32.77 | 31.37 |
| B | SJD (L=32) | 1031.2 | 42.99s | 2.32× | 2.37x | 30.78 | 32.82 | 31.31 |
| D | + **Ours** ($\pi_{MC}$) | 666.0 | 26.77s | 3.59× | 3.81x | 30.79 | 33.56 | 31.32 |
| D | + **Ours** ($\pi_{GS}$) | **652.3** | **26.44s** | **3.66×** | **3.86x** | 30.75 | 32.91 | 31.39 |
| B | SJD (L=64) | 1035.9 | 42.98s | 2.31× | 2.37x | 30.81 | 32.76 | 31.31 |
| D | + **Ours** ($\pi_{MC}$) | **567.7** | 24.41s | **4.21×** | 4.18x | 30.83 | 33.43 | 31.37 |
| D | + **Ours** ($\pi_{GS}$) | 568.0 | **24.24s** | 4.21x | **4.21x** | 30.90 | 32.80 | 31.37 |
| C | GSD (L=32,G=3) | 925.9 | 38.98s | 2.58× | 2.62x | 31.50 | 29.76 | 31.33 |
| C | GSD (L=32,G=10) | 701.4 | 29.13s | 3.40× | 3.50x | 33.21 | 26.78 | 31.25 |

latency breakdown of each component within single NFE step. As shown, *Transformer forward* dominates latency, while overhead of sampling operations, including our couplings, is under 5%. Moreover, since we define $p^{(t)}$ as the final distribution after all logit post-processing(i.e., top-k, CFG), these steps do not affect our exactness guarantee.

## 5. Experimental Results

In experiments section, we mainly focus on validating two aspects : (i) How much acceleration can we gain by applying our method atop SJD, (ii) Does our algorithm truly preserve generation quality, although we show it theoretically.

**Setup** Similar to original SJD paper, we mainly evaluate with Lumina-mGPT (Liu et al., 2024) for AR image generation. We also evaluate our method with the more SOTA AR Image model, Janus-Pro (Chen et al., 2025) and Lumina-mGPT-2 (Xin et al., 2025), to validate our method's generalization. Moreover, beyond image generation, we also evaluate with an AR video generation model, Cosmos-1-AR (Agarwal et al., 2025), which has significantly longer generation sequence Length. Detailed settings are in appendix.

**Metrics and Datasets** To evaluate the quality, we measured FID (Heusel et al., 2017) , which measures the distribution distance compared to reference and generated datasets, with IS (Barratt & Sharma, 2018) and CLIP score (Radford et al., 2021). To evaluate the speed, we measure the number of function evaluations (NFEs), which indicates the number of sequential forward steps for generation, and the real latency on 1x NVIDIA A100 device. We mainly use MS-COCO (Lin et al., 2014) and Parti-prompt (Yu et al., 2022) dataset for image generation and Real-Estate-10k for video generation. More details are in appendix.

**Baselines** We benchmarked our method against three baselines: (A) Autoregressive decoding (*Vanilla AR*), (B) Speculative Jacobi Decoding (SJD) (Teng et al., 2024) (C)

*Table 3.* Janus-Pro (7B) Results on MS-COCO dataset.

| | Config | NFE ($\downarrow$) | Latency(s)($\downarrow$) | FID ($\downarrow$) | IS ($\uparrow$) |
|---|---|---|---|---|---|
| A | *Vanilla AR* | 576 | 13.218 | 37.96 | 22.39 |
| B | SJD (L=16) | 319.93 | 10.213 | 37.96 | 22.25 |
| D | + **Ours**($\pi_{MC}$) | 190.21 | 6.345 | 37.46 | 22.19 |
| D | + **Ours**($\pi_{GS}$) | **189.99** | **6.336** | 37.13 | 22.53 |
| B | SJD (L=32) | 318.01 | 10.582 | 37.76 | 21.80 |
| D | + **Ours**($\pi_{MC}$) | 154.76 | 5.471 | 38.34 | 22.17 |
| D | + **Ours**($\pi_{GS}$) | **154.42** | **5.388** | 37.49 | 22.43 |

Grouped Speculative Decoding (GSD) (So et al., 2025), which is recently proposed training-free lossy SD methods for image generation and (D) Ours. We mainly compare our method against these baselines because they are training-free SD, thus enabling a fair comparison, and they demonstrate most superior performance. We also provide comparisons with training-based SD models in Appendix.

**Results** Table 1 presents our main results for AR image generation on Lumina-mGPT. As shown, our method (D) accelerates the *Vanilla AR* (A) by up to 4.2x and SJD (B) by 1.8x without compromising exactness guarantee, maintaining identical FID, IS and CLIP scores. Notably, while standard SJD fails to achieve meaningful speedup with an increased window size (L), our SCD shows higher acceleration as the window size grows, strongly suggesting that our coupling helps to stabilize SJD's convergence. Finally, (C) , lossy-SD method GSD, also significantly reduces the NFE, but results in a degradation of the FID and CLIP scores. Our method, in contrast, shows an even faster speedup than lossy GSD while maintaining quality exactness. In Table 3, we report results of Janus-Pro (7B). As shown, our method consistently accelerates the standard SJD process by up to 2.1x, achieving a final step compression of 3.7x. We also present results on Parti-prompt dataset and Lumina-mGPT-2 in Tab. 5, 6 and Fig. 10. As shown, our SCD still shows up to $4.4\times$ speedup compared to AR in various datasets and models.

*Table 4.* Video generation results on Cosmos1-AR-4B (Agarwal et al., 2025), on real-estate-10k dataset.

| Metric | Vanilla | L=16 | | | L=32 | | | L=64 | | | L=128 | | |
|---|---|---|---|---|---|---|---|---|---|---|---|---|---|
| | AR | SJD | $+ \pi_{\text{MC}}$ | $+ \pi_{\text{GS}}$ | SJD | $+ \pi_{\text{MC}}$ | $+ \pi_{\text{GS}}$ | SJD | $+ \pi_{\text{MC}}$ | $+ \pi_{\text{GS}}$ | SJD | $+ \pi_{\text{MC}}$ | $+ \pi_{\text{GS}}$ |
| NFE ($\downarrow$) | 7680 | 2272.8 | 1990.5 | **1940.6** | 1886.4 | 1293.7 | **1267.3** | 1802.3 | 835.9 | **810.7** | 1789.9 | 577.8 | **564.4** |
| Latency (s) ($\downarrow$) | 157.25 | 54.12 | 48.93 | **47.97** | 48.43 | 32.36 | **32.01** | 48.19 | 22.38 | **21.58** | 47.73 | 15.87 | **13.60** |
| FVD ($\downarrow$) | 156.9 | 157.1 | 159.3 | 154.8 | 153.2 | 155.8 | 153.6 | 163.6 | 155.8 | 152.9 | 158.3 | 157.8 | 152.4 |

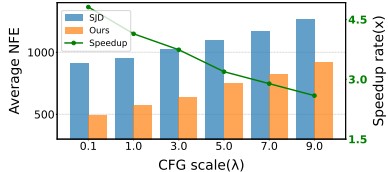
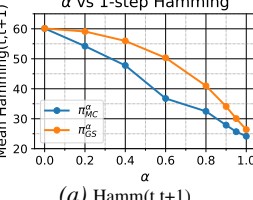
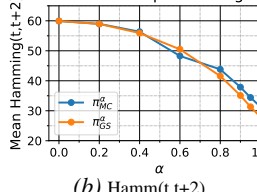
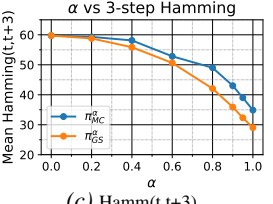

*Figure 6.* CFG $\lambda$ v.s. NFE on Lumina. Speedup denotes AR/Ours.

(a) Hamm(t,t+1)     (b) Hamm(t,t+2)     (c) Hamm(t,t+3)

*Figure 7.* Multi step behavior of draft tokens when our $\pi_{MC}$ and $\pi_{GS}$ applied.

In Table 4, We depict the results on AR video generation model, Cosmos-1-AR. Notably, our method achieves up to 13.6x actual acceleration with no loss in quality, outperforming other methods in significant margin. We believe this large acceleration mainly stems from the strong temporal redundancy between consecutive frames in video AR; thus making draft prediction relatively easy and thereby enabling the full utilization of longer draft window lengths.

### 5.1. Further Analysis

**Coupling Strength** To more precisely understand effect of coupling on convergence of SJD, we introduce a notion of *coupling strength* by interpolating between the independent distribution of SJD ($\pi_{SJD}$) and our coupling joint distribution ($\pi_{cpl}$). We define the $\alpha$-coupling by,

$$\pi_{\text{cpl}}^{\alpha}(x, y) = \alpha \cdot \pi_{\text{cpl}}(x, y) + (1 - \alpha) \cdot \pi_{\text{ind}}(x, y), \quad (4)$$

Thus $\alpha = 0$ recovers vanilla SJD, and $\alpha = 1$ corresponds to ours. We present validity proof and implementation in Appendix. In Fig.3b, we plot the mean token difference against the resulting NFE by changing $\alpha$. We observe a clear monotonic trend: as $\alpha$ increases, both the mean Hamming distance and the NFE consistently decrease, and this improvement continues up to $\alpha \approx 1$, strongly confirming our hypothesis from Sec. 3 that increasing the collision probability in token space directly enhances context stability and reduces the total NFEs required for generation.

**Multi-Step Behavior** In Fig. 7, we visualize the Hamming distance (mean token difference) between $N$-step iterations, $\text{Hamm}(t, t+N)$, when our couplers are applied. As shown, for the 1-step case, $\pi_{MC}$ consistently has a smaller distance than $\pi_{GS}$, aligning with our theory. However, for multi-step ($N = 2, 3$), this relationship is **reversed** in the high coupling-strength regime. This occurs because, while $\pi_{MC}$ is optimal for maximizing the 1-step collision probability, there is no non-trivial bound on its multi-step, whereas $\pi_{GS}$ have a lower bound of $(1 - \mathcal{D}_{TV})/(1 + \mathcal{D}_{TV})$ between any pair of multi-step iterations, as shown in Theorem 4.5. We interpret this as $\pi_{GS}$ having better *long-range stability*

than $\pi_{MC}$, and this can be advantageous in tasks where draft prediction is relatively easy, such as video AR(Tab. 4) which have high temporal similarity between frames or low-resolution image AR, such as Janus (Tab. 3). In these cases, retaining early draft tokens can yields better convergence (lower NFEs) than continuously updating them for marginal information improvement.

**Effect on CFG** AR vision models typically rely on CFG (Ho & Salimans, 2022) to control prompt alignment and fidelity, using scale around 3∼5. Specifically, the samples are generated from a mixed logit: $(1 + \lambda) \cdot c - \lambda \cdot u$, where logit $c$ generated with prompt and $u$ generated with masked prompt. As shown in Fig. 6, as $\lambda$ increases, speedup slightly decreased because the final logit becomes sharper. However, our method consistently outperforms SJD by large margin in practical range of scale $\lambda$.

**Qualitative Results** While we have theoretically and quantitatively show the losslessness of our method, we also performed a qualitative comparison to visualize that our method does not degrade generation quality. As shown in Fig. 2, our method yields outputs that are visually indistinguishable from the AR model while achieving the ∼4× acceleration. We provide more visualizations in Appendix.

## 6. Conclusion

In this paper, we identify and resolve a critical performance bottleneck in the recently proposed Speculative Jacobi Decoding (SJD). Specifically, we find that its acceptance rate is significantly limited by draft context instabilities, arising from its independent sampling process. To solve this, we propose Speculative Coupled Decoding (SCD), which use *Coupling* to replace independent sampling and stabilize the Jacobi iteration trajectory by increasing the probability of sampling identical tokens, transferring distributional similarity to the realized token space. As a result, we show that this simple tweak can remarkably enhance the speedup in visual AR, achieving up to 4.2× in image and 13.6× in video, all without any quality degradation and additional training.

## Acknowledgment

This work was supported by IITP and NRF grant funded by the Korea government(MSIT) (No. RS-2019-II191906, RS-2024-00457882, RS-2026-25490269) and Samsung Research Global AI Center.

## Impact Statement

This paper presents work to accelerate the inference speed of generative models. We believe there are no significant societal concerns to be specifically highlighted here

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

# A. Appendix

# B. Proofs

### B.1. Proof of Proposition 2.1

*Proof.* We will first check that $\text{MRS}(\cdot)$ returns $Y \sim P$ with input $X \sim Q$. Let the acceptance probability $min(1, p(x)/q(x)) = \alpha(x)$. Then, we can re-write the p.d.f of R.V $Y$, $y(x)$ as follows

$$y(x) = \alpha(x) \cdot q(x) + (1 - \sum_{x' \in V} \alpha(x') \cdot q(x'))r(x) \tag{5}$$

where $r(x)$ is residual distribution $r(x) = norm(max(0, p(x) - q(x)) = \frac{max(0, p(x) - q(x))}{\sum_{x' \in V} max(0, p(x') - q(x'))}$. We can rewrite the left term as :

$$\alpha(x) \cdot q(x) = min(1, p(x)/q(x)) \cdot q(x) = min(q(x), p(x)) \tag{6}$$

also with the right term :

$$(1 - \sum_{x' \in V} \alpha(x') \cdot q(x'))r(x) = (1 - \sum_{x' \in V} min(q(x'), p(x'))) \frac{max(0, p(x) - q(x))}{\sum_{x' \in V} max(0, p(x') - q(x'))} \tag{7}$$

$$= (1 - \sum_{x' \in V} min(q(x'), p(x'))) \frac{p(x) - min(p(x), q(x))}{\sum_{x' \in V} p(x') - min(p(x'), q(x'))} \tag{8}$$

$$= (1 - \sum_{x' \in V} min(q(x'), p(x'))) \frac{p(x) - min(p(x), q(x))}{1 - \sum_{x' \in V} min(p(x'), q(x'))} \tag{9}$$

$$= p(x) - min(p(x), q(x)). \tag{10}$$

So, adding two terms becomes $p(x)$, the target distribution, as desired.

Now, we will check the acceptance rate :

$$\mathbb{E}_{x' \sim q(x)} min(1, \frac{p(x')}{q(x')}) = \sum_{x' \in V} q(x') \cdot min(1, \frac{p(x')}{q(x')}) = \sum_{x' \in V} min(p(x'), q(x')) \tag{11}$$

$$= 1/2 \sum_{x' \in V} p(x') + q(x') - |p(x') - q(x')| = 1 - \frac{1}{2} \sum_{x' \in V} |p(x') - q(x')| \tag{12}$$

which is $1 - \mathcal{D}_{TV}(p, q)$.

### B.2. Proof of Proposition 3.3

To compute value of $C_{SJD}$, let $p(x) = p_i^{(t)}(x)$ and $q(x) = p_i^{(t-1)}(x)$ for simplicity.

$$C_{SJD}(p, q) \equiv Pr[X^{(t)} = X^{(t-1)}] \tag{13}$$

$$= \sum_{x \in \mathcal{V}} Pr[X^{(t)} = x, X^{(t-1)} = x] \tag{14}$$

$$= \sum_{x \in \mathcal{V}} Pr[X^{(t)} = x] \cdot Pr[X^{(t-1)} = x] \qquad \text{(by independence)} \tag{15}$$

$$= \sum_{x \in \mathcal{V}} p(x)q(x) \tag{16}$$

Now, we will derive it's upper bound as follows :

$$(C_{SJD}(p,q))^2 = \left(\sum_x p(x)q(x)\right)^2 \tag{17}$$

$$\leq \left(\sum_x p(x)^2\right)\left(\sum_x q(x)^2\right) \qquad \text{(by Cauchy-Schwarz)} \tag{18}$$

Since Renyi-2 entropy is, by definition, $H_2(p) = -\log\left(\sum_x p(x)^2\right) \implies \sum_x p(x)^2 = e^{-H_2(p)}$,

Hence,

$$(C_{SJD}(p,q))^2 \leq e^{-H_2(p)} \cdot e^{-H_2(q)} = e^{-(H_2(p)+H_2(q))} \tag{19}$$

$$\implies C_{SJD}(p,q) \leq e^{-\frac{1}{2}(H_2(p)+H_2(q))} \tag{20}$$

So we can check that independence collision probability is exponentially restricted by their Renyi-2 entropy, regardless of how they are close to each other.

### B.3. Proof of Theorem 4.2

*Proof sketch.* The theoretical correctness of our approach is based on the marginalization property of the couplings. The standard SJD framework requires that the draft token, which we denote $X_i^{(t)}$, be sampled from the draft distribution $p_i^{(t)}$. If sample $X_i^{(t)}$ follows it's distribution, then correctness of SD framework is guaranteed by Proposition 2.1. According to Definition 4.1, when we sample a pair $(X_i^{(t)}, X_i^{(t-1)}) \sim \pi(\cdot, \cdot)$ from any valid coupling $\pi \in \Pi_i^{(t)}$, the marginal distribution of the variable $X_i^{(t)}$ is precisely $p_i^{(t)}$. Thus, using the $X_i^{(t)}$ component from the sampled pair is probabilistically identical to sampling a token directly from $p_i^{(t)}$. Since this modification preserves the required sampling distribution for the draft token at each step, the final output distribution of the algorithm is guaranteed to match that of the base model.

### B.4. Proof of Theorem 4.4

We will formally check that $\text{MRS}(\cdot)$ satisfies the definition of Coupling. Let the joint distribution of this $\text{MRS}(\cdot)$ process $f(x,y)$ is

$$f(x,y) = q(x)(\alpha(x)\delta_x(y) + (1-\alpha(x))r(y)) \tag{21}$$

where $\delta_x(y)$ is kronecker-delta symbol.

Let $\alpha(x)$, $r(x)$ is same as we defined on proof of Proposition 2.1. Then for $p(y)$,

$$\sum_{x \in V} f(x,y) = a(y) \cdot q(y) + (1 - \sum_{x \in V} \alpha(x)q(x))r(y) = p(y) \tag{22}$$

which directly came out from proof of Proposition 2.1.

Then next, for $q(x)$,

$$\sum_{y \in V} f(x,y) = q(x)\alpha(x)\sum_{y \in V}\delta_x(y) + q(x)(1-\alpha(x))\sum_{x \in V} r(y) \tag{23}$$

$$= q(x)\alpha(x) + q(x)(1-\alpha(x)) = q(x) \tag{24}$$

So it satisfies the definition of Coupling.

For the coupling cost optimality, it is well studied that any coupling can not have cost greater than $1 - \mathcal{D}_{TV}(P,Q)$ (Lindvall inequality). See (Lindvall, 2002; Bavarian et al., 2016).

### B.5. Proof of Theorem 4.5

The coupling validity of $\pi_{GS}$ can be easily shown based on well-known Gumbel-Max Trick (Gumbel, 1954), where this trick known to yield output that follows input categorical distribution. To show its lower bound on coupling cost, we start with notation of exponential race :

$$X = \arg\min_i \frac{E_i}{P_i}, \qquad Y = \arg\min_i \frac{E_i}{Q_i}.$$

, where $E_i \sim \text{Exp}(1)$ is Exponential noise. Because $Pr[X = Y] = \sum_k Pr[X = Y = k]$, We will first derive $Pr[X = Y = k]$ for fixed $k$ and will integrate it. For fixed $k \in \mathcal{V}$ and condition on $E_k = a$, $\{X = Y = k\}$ requires that for every $j \neq k$,

$$E_j \geq a \cdot \max\left(\frac{P_j}{P_k}, \frac{Q_j}{Q_k}\right).$$

Since $\Pr[E_j \geq a] = e^{-a}$ and the $E_j$ are independent with $P$ and $Q$,

$$\Pr[X = Y = k \mid E_k = e] = \exp\left(-a \sum_{j \neq k} \max\left(\frac{P_j}{P_k}, \frac{Q_j}{Q_k}\right)\right).$$

Integrating over $E_k$ yields

$$\Pr[X = Y = k] = \int_0^\infty e^{-a} \cdot \exp\left(-a \sum_{j \neq k} \max\left(\frac{P_j}{P_k}, \frac{Q_j}{Q_k}\right)\right) da = \frac{1}{\sum_{j \in \mathcal{V}} \max\left(\frac{P_j}{P_k}, \frac{Q_j}{Q_k}\right)}. \tag{6}$$

Let $m_k := \min(P_k, Q_k)$ and $M_j := \max(P_j, Q_j)$. Since $P_k, Q_k \geq m_k$, for each $j$,

$$\max\left(\frac{P_j}{P_k}, \frac{Q_j}{Q_k}\right) \leq \frac{M_j}{m_k}.$$

Therefore, from Eq. (6),

$$\Pr[X = Y = k] \geq \frac{m_k}{\sum_j M_j}. \tag{7}$$

Summing over $k$ gives

$$\Pr[X = Y] \geq \frac{\sum_k \min(P_k, Q_k)}{\sum_j \max(P_j, Q_j)}. \tag{8}$$

Finally,

$$\sum_k \min(P_k, Q_k) = 1 - D_{\text{TV}}(P, Q), \qquad \sum_k \max(P_k, Q_k) = 1 + D_{\text{TV}}(P, Q),$$

so

$$\Pr[X = Y] \geq \frac{1 - D_{\text{TV}}(P, Q)}{1 + D_{\text{TV}}(P, Q)}.$$

Please see (Bavarian et al., 2016) also.

### B.6. Proof of $\alpha$-Coupling

For Sec 5.1, we define $\alpha$-Coupling as follows :

$$\pi_{\text{cpl}}^\alpha(x, y) = \alpha \cdot \pi_{\text{cpl}}(x, y) + (1 - \alpha) \cdot \pi_{\text{ind}}(x, y), \tag{25}$$

where $0 \leq \alpha \leq 1$. We will now show this linear-interpolation of two coupling is still valid coupling.

To check marginal property, we can rewrite it as :

$$\sum_{y \in V} \pi_{cpl}^{a}(x, y) = \sum_{y \in V} [\alpha \cdot \pi_{cpl}(x, y) + (1 - \alpha) \cdot \pi_{ind}(x, y)] \tag{26}$$

$$= \alpha \cdot \sum_{y \in V} \pi_{cpl(x,y)} + (1 - \alpha) \cdot \sum_{y \in V} \pi_{ind}(x, y) \tag{27}$$

$$= \alpha \cdot p(x) + (1 - \alpha) \cdot p(x) = p(x) \tag{28}$$

Same procedure can be applied to $q(y)$.

**Implementation** To implement linear interpolation of two probability distribution, we can simply sample R.V from Bernoulli distribution, and pick sample with that probability. For example :

---

**Algorithm 5** $\alpha$-Coupling Implementation

---

**Input**: Coupling strength $\alpha$, Two valid coupling $\pi_a$, $\pi_b$.
**Output**: Random Variable X.

1: $u \sim \mathcal{U}(0, 1)$
2: **if** $u \leq \alpha$ **then**
3:     $X, Y \leftarrow \pi_a(x, y)$
4: **else**
5:     $X, Y \leftarrow \pi_b(x, y)$
6: **end if**
7: **return** $X$

---

## C. Related Works

**Unified Multimodal Models.** Recently, Unified Multimodal Models (Team, 2024; Deng et al., 2025; Hurst et al., 2024), which can process data from multiple modalities such as text, images, and audio for both input and output within a single model, have gained significant attention. The advantage of this paradigm stems from the discovery that models trained on multiple data domains simultaneously exhibit superior performance across a range of tasks compared to single-modality models. This includes enhanced understanding, generation (Chen et al., 2025), complex world reasoning (Hurst et al., 2024), instruction following, and iterative editing (Bai et al., 2023).

**Autoregressive Models in Vision** Visual generation using an autoregressive (AR) (Team, 2024) approach is a promising method for implementing Unified Multimodal Models. An AR vision model primarily consists of two key components: a Vector Quantizer (Van Den Oord et al., 2017) and a Transformer model (Brown et al., 2020). The vector quantizer divides an image into patches of a specified size and maps each patch to a discrete code from a predefined codebook. This process effectively performs both downsampling and tokenization of the image. Subsequently, similar to autoregressive text generation, a Transformer model is trained to predict these visual token IDs autoregressively. This paradigm enables the learning and inference of diverse data types under a single, unified framework of AR modeling, naturally facilitating stable training, deployment, and capabilities such as in-context learning (Hurst et al., 2024), editing (Liu et al., 2024), and reasoning (Zhao et al., 2025).

**Speculative Decoding** Speculative Decoding (SD) was first proposed by (Leviathan et al., 2023; Chen et al., 2023) to accelerate the inference speed of Large Language Models (LLMs) without compromising performance by generating multiple tokens at once. Later, (Sun et al., 2023) established a connection between speculative sampling and optimal transport, proving that the token-level acceptance scheme is theoretically optimal for individual tokens. More recently, (Sun et al., 2024b) showed that token-level acceptance is not globally optimal and that the block-wise acceptance approach is the theoretically optimal form of speculative decoding. As the theoretical optimality has been established, the recent research trend in SD has focused on designing better draft models (?Brown et al., 2024; Cai et al., 2024) or exploring methods that trade speed for a slight degradation in quality (Bachmann et al., 2025; So et al., 2025).

**Parallel Decoding** Parallel decoding, or fixed-point iteration $X \leftarrow F(X)$, is a widely used technique for rapidly finding the solution to a specific system, from scientific computing for accelerating the solution of differential equations (Berinde, 2004) to, more recently, fast sampling of diffusion models (Shih et al., 2023). Building on this concept, (Song et al., 2021) first

proposed using fixed-point iteration to accelerate the sequential computation of neural networks. Based on the observation that this method guarantees the same result as sequential computation and always at least as faster than sequential when assuming fully parallelization model. Our method can be framed as a novel methodology for accelerating the convergence speed of fixed-point (jacobi) iteration for sequential sampling that operates based on a probabilistic process within a discrete space.

## D. Limitations

Since our method relies on parallel computation, the overhead of parallel operations may become significant if the window size or batch size becomes too large, potentially failing to achieve practical speedups, especially in scenarios where the parallel-computation unit is limited. However, this is a structural limitation shared by all Speculative Decoding (SD) methods, and we predict that these limitations will gradually disappear with the advancement of modern hardware.

## E. Experimental Details

### E.1. Image Generation

**Lumina mGPT:** For Lumina-mGPT (Liu et al., 2024) , we use the standard 7B model and experiment with resolution of $768\times768$. In all experiments, we follow the default settings of vanilla model, temperature $\tau = 1$ and Top-K sampling with $K = 2000$ and guidance scale of $\lambda = 3.0$. We used pytorch 2.3 (Paszke et al., 2019) for the main comparison. For quality evaluation, we generate 5000 images for each MS-COCO 2017 (val) (Lin et al., 2014) prompt and compute FID, IS, CLIP-Score with reference dataset.

**Janus Pro :** For Janus-Pro (Chen et al., 2025), we use 7B model to generate images at a resolution of $384 \times 384$. Following the setup of the vanilla Janus-Pro 7B model, $24 \times 24$ of image tokens are generated with a downsampling size of 16. For sampling, we follows vanilla setting that guidance scale of 5.0 and temperature of 1.0. We also adopted a Top-$K$ logits processor with $K = 1000$. For evaluation, we generate three images for each MS-COCO (val) prompt with different seeds ($5000\times3$) and reported the mean values of the FID, IS, and CLIP score across the seeds.

### E.2. Video Generation

**Cosmos1-autoregressive.** We evaluate our method on the *Cosmos-1.0-Autoregressive-4B* video AR model (Agarwal et al., 2025) using a curated subset of 150 clips from the *real-estate-10k* dataset (Zhou et al., 2018). For each clip, we provide a 9-frame context to the model and autoregressively generate the next 24 frames, yielding 33-frame sequences in total (9 observed + 24 predicted). Unless otherwise noted, decoding uses nucleus (top-$p$) sampling with $p = 0.8$ and temperature 1.0.

We compare three decoders: (A) vanilla AR, (B) Speculative Jacobi Decoding (SJD), and (D) our SCD on top of SJD. For SJD-based methods we sweep the parallel verification window $L \in \{16, 32, 64, 128\}$. Speed is reported as (i) **NFE**—the number of sequential target-model evaluations—and (ii) end-to-end wall-clock **Latency** (seconds) measured on a single RTX6000ADA. Quality is measured by **FVD (Fréchet Video Distance)** (Unterthiner et al., 2019), computed between the generated frames and the corresponding ground-truth future frames of each clip.

## F. Algorithms

We provide complete pseudo code of actual efficient implementation version of our SCD in Algorithm 6 (Maximal Coupling) and Algorithm 7 (Gumbel Coupling). As shown, they are computationally almost identical with original SJD, thus having almost zero-overhead in terms of both computation and memory.

---

**Algorithm 6** Speculative Coupled Decoding ($\pi_{MC}$)

---

**Require:** AR Model $p_\theta$, draft Length $L$, Max Sequence $N$

1: $p_t^i \leftarrow$ `Random()`
2: $X_i^t \sim p_i^t \quad \forall i, t$           $\triangleleft$ Initialize

3: **While** $i < N$
4:    **parallel for** $j = i$ to $i + L$ :       $\triangleleft$ Evaluate
5:      $p_j^{t+1} \leftarrow p_\theta(\cdot \mid X_{0:j-1}^t)$

6:    $Accept \leftarrow i + L$

7:    **parallel for** $j = i$ to $i + L$ :     $\triangleleft$ Verify & Drafting with Vectorized `MRS`
8:      $k, X_j^{t+1} \leftarrow$ `MRS`$(p_j^{t+1}, p_j^t, X_j^t)$

9:      **if** $k = 0$ and $Accept = i + L$ :     $\triangleleft$ Find first rejection index
10:         $Accept \leftarrow j$

11:    $i \leftarrow Accept + 1$ , $t \leftarrow t + 1$
12: **Return** $X$

---

---

**Algorithm 7** Speculative Coupled Decoding ($\pi_{GS}$)

---

**Require:** AR Model $p_\theta$, draft Length $L$, Max Sequence $N$

1: $p_t^i \leftarrow$ `Random()`           $\triangleleft$ Initialize state
2: $X_i^t \sim p_i^t \quad \forall i, t$           $\triangleleft$ //Initialize

3: **While** $i < N$
4:    **parallel for** $j = i$ to $i + L$ :       $\triangleleft$ Drafting
5:      $G_j \leftarrow$ `SampleGumbelNoiseUsingHash`$(j)$    $\triangleleft G_j$.shape = $[\mathcal{V}]$
6:      $X_j \leftarrow \text{argmax}_{i \in \mathcal{V}}(\log(p_{j_i}^t) + G_{j_i})$     $\triangleleft$ Fixed noise sampling

7:    **parallel for** $j = i$ to $i + L$ :       $\triangleleft$ Evaluate
8:      $p_j^{t+1} \leftarrow p_\theta(\cdot \mid X_{<j}^t)$

9:    **for** $j = i$ to $i + L$ :       $\triangleleft$ Verify
10:      $k, X_j^{t+1} \leftarrow$ `MRS`$(p_j^{t+1}, p_j^t, X_j^t)$, **if** $k = 0$ : **break**

11:    $i \leftarrow j + 1$, $t \leftarrow t + 1$
12: **Return** $X$

---

## G. More Visualization

In this section, we provide further details about the visualization settings and discuss our findings based on both quantitative and qualitative results. For image generation, we employed prompts covering diverse categories such as humans, animals, landscapes, close-up shots, fantasy, and paintings. In particular, we included prompts designed to capture physical phenomena such as reflections and waves. We also incorporated descriptors explicitly indicating high-quality imagery (e.g., 8K, sharp focus) to encourage the generation of fine-detailed, realistic images.

As shown in Figs. 8, 9, 10, 11, we observed that our method produced images closely resembling those of the vanilla AR model while achieved more than a $4\times$ reduction in NFE in image generation and $13\times$ in video generation. Moreover, our model was able to generate diverse categories of images, including physical phenomena like reflections and waves, under both the maximal coupling and the Gumbel coupling.

*Table 5.* Evaluation results of AR Image generation model, Lumina-mGPT, on Parti-Prompt.

| Configuration | | NFE (↓) | Latency (↓) | Acceleration (↑) | | CLIP-Score (↑) |
|---|---|---|---|---|---|---|
| | | | (A100) | NFE | Latency | |
| A | *Vanilla AR* | 2392 | 112.29 | 1.00× | 1.00× | 32.091 |
| B | SJD (L=16) | 1035.3 | 42.07s | 2.31x | 2.67x | 32.11 |
| D | + **Ours** ($\pi_{MC}$) | **817.29** | **32.86s** | **2.92x** | **3.42x** | 32.12 |
| D | + **Ours** ($\pi_{GS}$) | 820.12 | 32.92s | 2.91x | 3.41x | 32.07 |
| B | SJD (L=32) | 1038.2 | 43.27s | 2.30x | 2.60x | 32.087 |
| D | + **Ours** ($\pi_{MC}$) | **647.08** | **26.01s** | **3.69x** | **4.32x** | 32.09 |
| D | + **Ours** ($\pi_{GS}$) | 649.54 | 26.33s | 3.68x | 4.26x | 32.102 |
| B | SJD (L=64) | 1036.2 | 42.99s | 2.31x | 2.61x | 32.095 |
| D | + **Ours** ($\pi_{MC}$) | **548.26** | 23.58s | **4.36x** | 4.76x | 32.113 |
| D | + **Ours** ($\pi_{GS}$) | 548.75 | **23.42s** | **4.36x** | **4.79x** | 32.089 |
| C | GSD (G=50) | 636.75 | 25.24s | 3.76x | 4.45x | 32.075 |

| Config | | NFE (↓) | Latency(s)(↓) | CLIP-Score (↑) |
|---|---|---|---|---|
| A | *Vanilla AR* | 576 | 16.84s | 28.98 |
| B | SJD (L=16) | 312.00 | 9.96s | 29.02 |
| D | + **Ours**($\pi_{MC}$) | **186.48** | **6.22s** | 28.99 |
| D | + **Ours**($\pi_{GS}$) | 189.99 | 6.336 | 29.03 |
| B | SJD (L=32) | 308.17 | 10.25s | 28.97 |
| D | + **Ours**($\pi_{MC}$) | **149.53** | 5.471 | 28.95 |
| D | + **Ours**($\pi_{GS}$) | 154.42 | **5.388** | 29.13 |

*Table 6.* Evaluation results of AR Image generation model, Janus-Pro, on Parti-Prompt.

# H. More Dataset

To validate the generalization capabilities of our methods across various datasets, we conducted a new experiment using the Parti-Prompt (Yu et al., 2022) dataset for text-to-image generation. Specifically, we evaluated generation quality using the CLIP score, and utilized a set of 1,600 distinct real-world text prompts for evaluation; all other experimental settings remain identical to the MS-COCO evaluation described in the main paper.

Tables 5 and 6 present the results on the Parti-Prompt dataset. As shown, our methods exhibit acceleration rates of approximately $\sim 4\times$, which are comparable to those observed on the MS-COCO dataset, demonstrating the strong generalization capabilities of our proposed methods.

# I. Comparison with Training-based SD

While we mainly compare our method with training-free SD for fair comparison, we also present quantitative comparison result with recent training-based SD, EAGLE (Brown et al., 2024) and LANTERN (Jang et al., 2024), in Tab. 7. We use Llamagen XL (Stage 2) (Sun et al., 2024a) using 500 MS-COCO prompts. As shown, our SCD surpasses all baselines by a large margin, outperforming the training-based SD EAGLE and even the lossy-training-based-SD (LANTERN), while maintaining training-free and lossless-ness constraints.

# J. Scaling with Sequence Length

To examine the scaling behavior of our method, we evaluate Lumina-mGPT (Liu et al., 2024) under different generation resolutions. Since higher resolutions induce longer generation sequences, this setting allows us to verify whether the

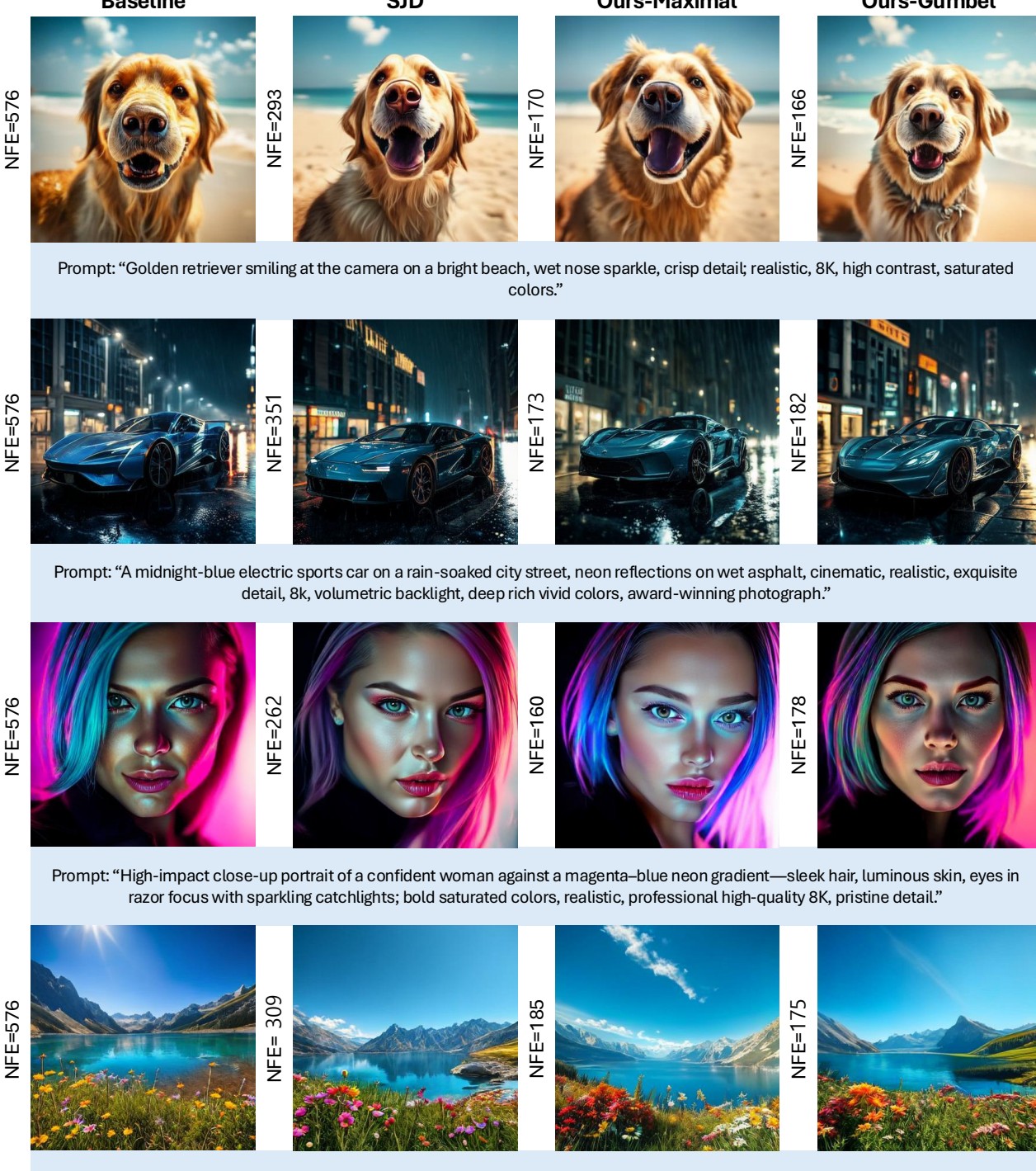

*Figure 8.* Qualitative comparison on Janus-Pro 7B

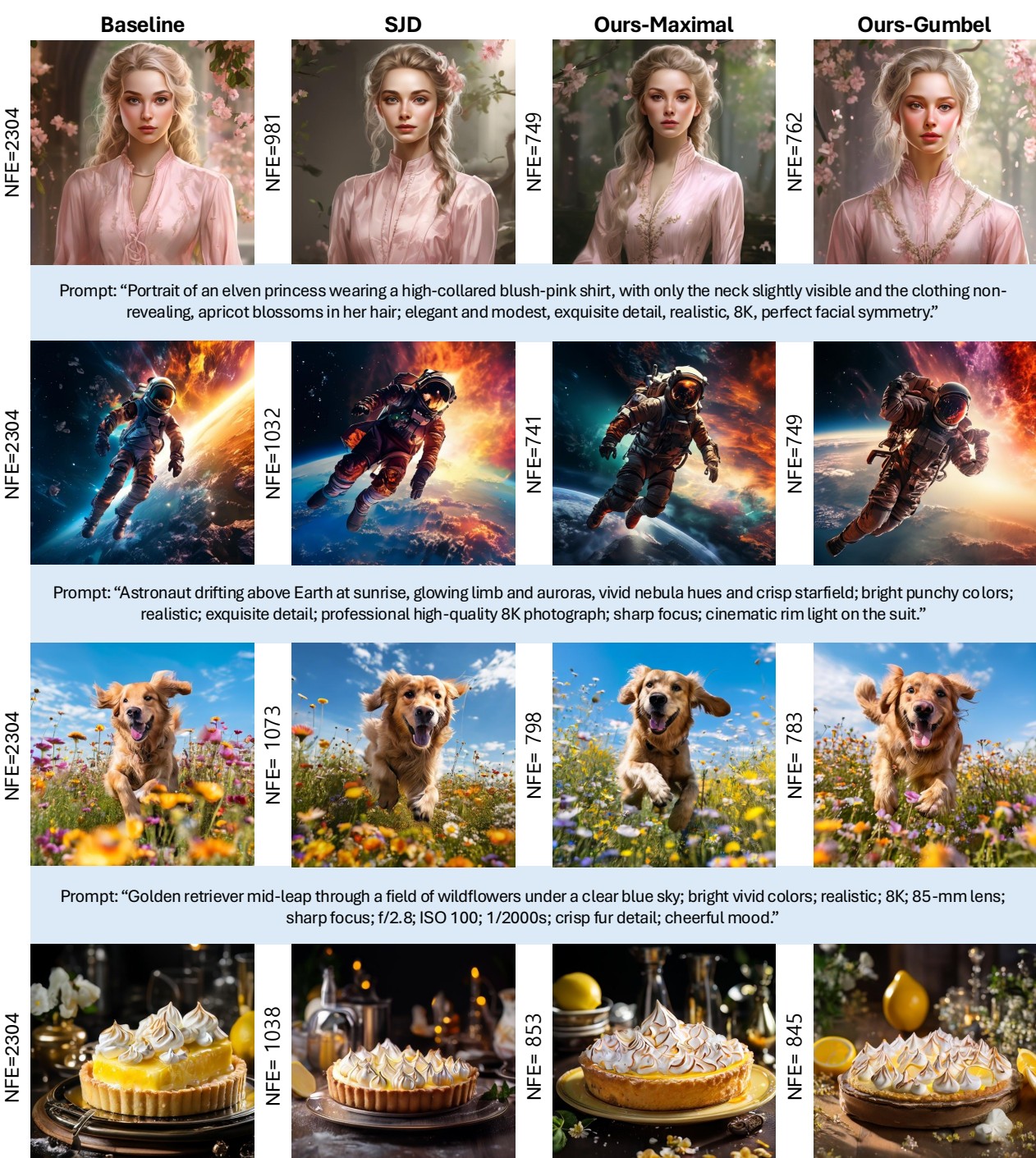

*Figure 9.* Qualitative comparison on Lumina-mGPT (1.0)

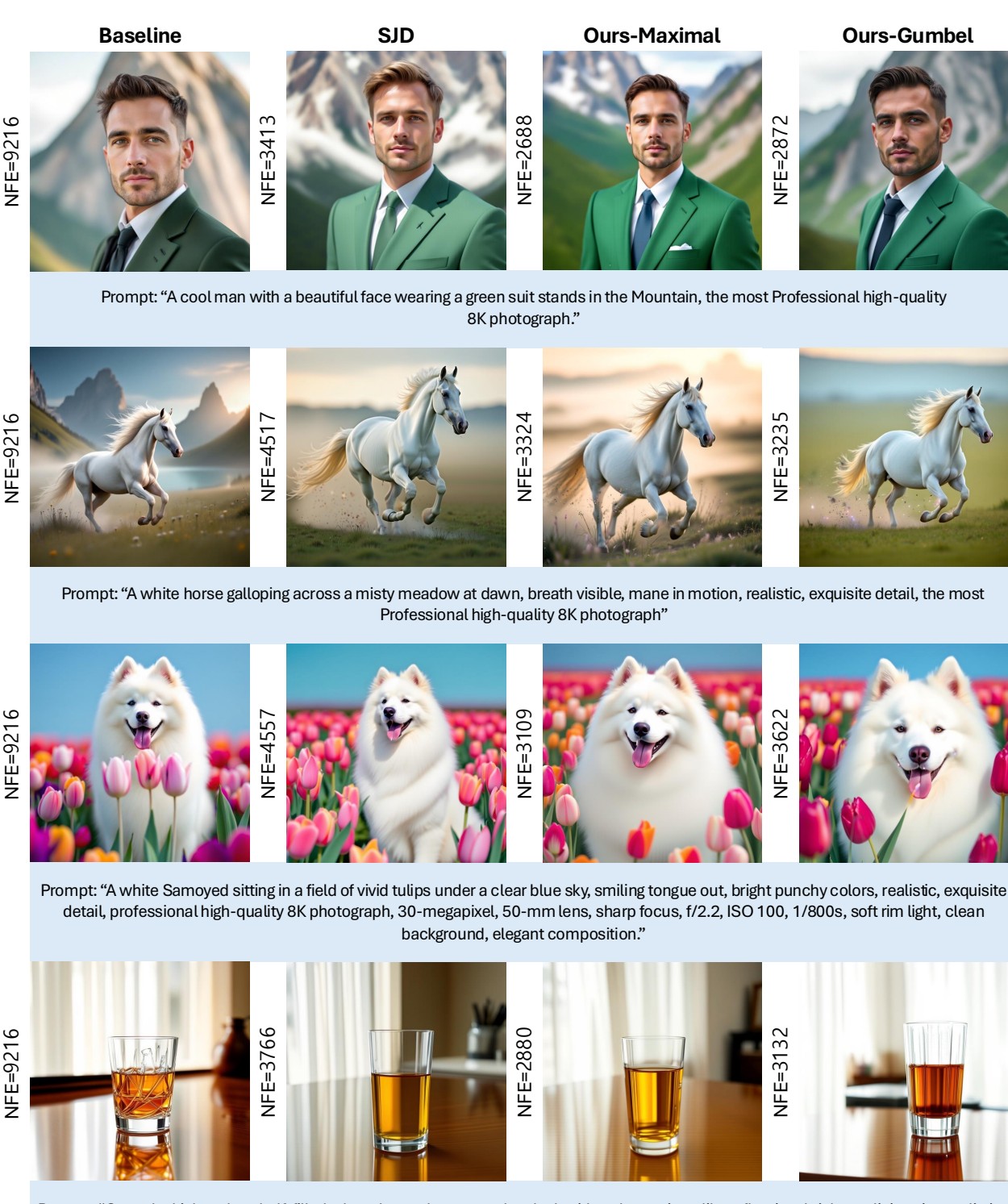

*Figure 10.* Qualitative comparison on Lumina-mGPT 2.0

| Method | Training-Free? | Lossless? | NFE ($\downarrow$) |
|---|---|---|---|
| *Vanilla AR* | - | - | 1024. |
| EAGLE (Brown et al., 2024) | ✗ | ✓ | 853.30 |
| LANTERN ($\delta = 0.1$) (Jang et al., 2024) | ✗ | ✗ | 585.14 |
| SJD (Teng et al., 2024) | ✓ | ✓ | 632.49 |
| **Ours($\pi_{MC}$)** | ✓ | ✓ | **515.72** |

*Table 7.* Comparison with Training-based SD

| Resolution | AR | SJD (L=16) | Ours (L=16) | SJD (L=32) | Ours (L=32) | SJD (L=64) | Ours (L=64) |
|---|---|---|---|---|---|---|---|
| $512\times512$ | 1131 | 421.9 | **357.2** | 443.5 | **278.5** | 443.7 | **211.5** |
| $768\times768$ | 2390 | 1058.6 | **814.5** | 1031.2 | **666.0** | 1035.9 | **567.7** |
| $1024\times1024$ | 4237 | 1765.9 | **1284.1** | 1771.8 | **988.7** | 1735.5 | **785.3** |

*Table 8.* Scaling results of Lumina-mGPT under different generation resolutions. We report NFE, where lower is better.

acceleration gain of our method persists as sequence length increases.

As shown in Table 8, our method consistently reduces NFE across all tested resolutions and all draft lengths. Notably, the relative gain remains substantial even at $1024\times1024$, indicating that the advantage of our method persists as the sequence length increases.

## K. Integration with Grouped Speculative Decoding

Grouped Speculative Decoding (GSD) (So et al., 2025) is applied at the verification stage of speculative decoding, whereas our SCD is applied at the drafting stage. Therefore, the two approaches are complementary and can be combined synergistically. We evaluate this combination on both image and video generation tasks.

As shown in Tables 9 and 10, combining GSD with SCD yields additional acceleration over SCD alone. On Lumina-mGPT, SCD with GSD reduces NFE from 657.2 to 298.2 when using $G = 10$. Similarly, on Cosmos1-AR, SCD with GSD further reduces NFE from 565 to 444. These results suggest that our method can be effectively combined with lossy acceleration schemes, with only a small degradation in generation quality.

## L. Applicability to Other Modalities

We expect our method to apply broadly to other modalities that use Vector Quantization (VQ) (Van Den Oord et al., 2017) for tokenization. Since VQ discretizes continuous signals into codebook entries, different entries may share overlapping semantics, leading to high-entropy next-token distributions. Our SCD is particularly effective in this regime, where high entropy causes draft-token instability through low collision probabilities.

To demonstrate this, we conduct an additional experiment on a text-to-speech (TTS) task using the audio LLM CosyVoice2 (Du et al., 2024) on the LibriTTS dataset. The results are summarized in Table 11.

As shown in Table 11, SCD consistently accelerates decoding for audio autoregressive generation, achieving up to $1.39\times$ speedup while maintaining comparable WER. These results indicate that our method is not limited to image or video generation, and can potentially be extended to other VQ-based autoregressive models, such as those for 3D or motion generation.

| Method | FID ($\downarrow$) | NFE ($\downarrow$) |
|---|---|---|
| AR | 30.79 | 2390 |
| SJD (L=32) | 30.78 | 1031.2 |
| SJD (L=32) + GSD (G=3) | 31.50 | 925.9 |
| SJD (L=32) + GSD (G=10) | 33.21 | 701.4 |
| **SCD (L=32)** | 30.78 | **657.2** |
| **SCD (L=32) + GSD (G=3)** | 30.99 | **416.65** |
| **SCD (L=32) + GSD (G=10)** | 32.91 | **298.2** |

*Table 9.* Integration of SCD with GSD on Lumina-mGPT for image generation.

| Method | FVD ($\downarrow$) | NFE ($\downarrow$) |
|---|---|---|
| AR | 170.3 | 7680 |
| SJD (L=128) | 169.8 | 1786 |
| SJD (L=128) + GSD (G=3) | 165.5 | 1030 |
| SJD (L=128) + GSD (G=30) | 171.0 | 926 |
| **SCD (L=128)** | 168.2 | **565** |
| **SCD (L=128) + GSD (G=3)** | 166.1 | **455** |
| **SCD (L=128) + GSD (G=30)** | 172.7 | **444** |

*Table 10.* Integration of SCD with GSD on Cosmos1-AR for video generation.

| Method | Speedup ($\uparrow$) | WER ($\downarrow$) |
|---|---|---|
| AR | 1.00$\times$ | 3.49% |
| **SCD (L=8)** | 1.32$\times$ | 3.30% |
| **SCD (L=16)** | 1.33$\times$ | 3.68% |
| **SCD (L=32)** | 1.39$\times$ | 3.30% |

*Table 11.* Results on a text-to-speech task using CosyVoice2 on LibriTTS.

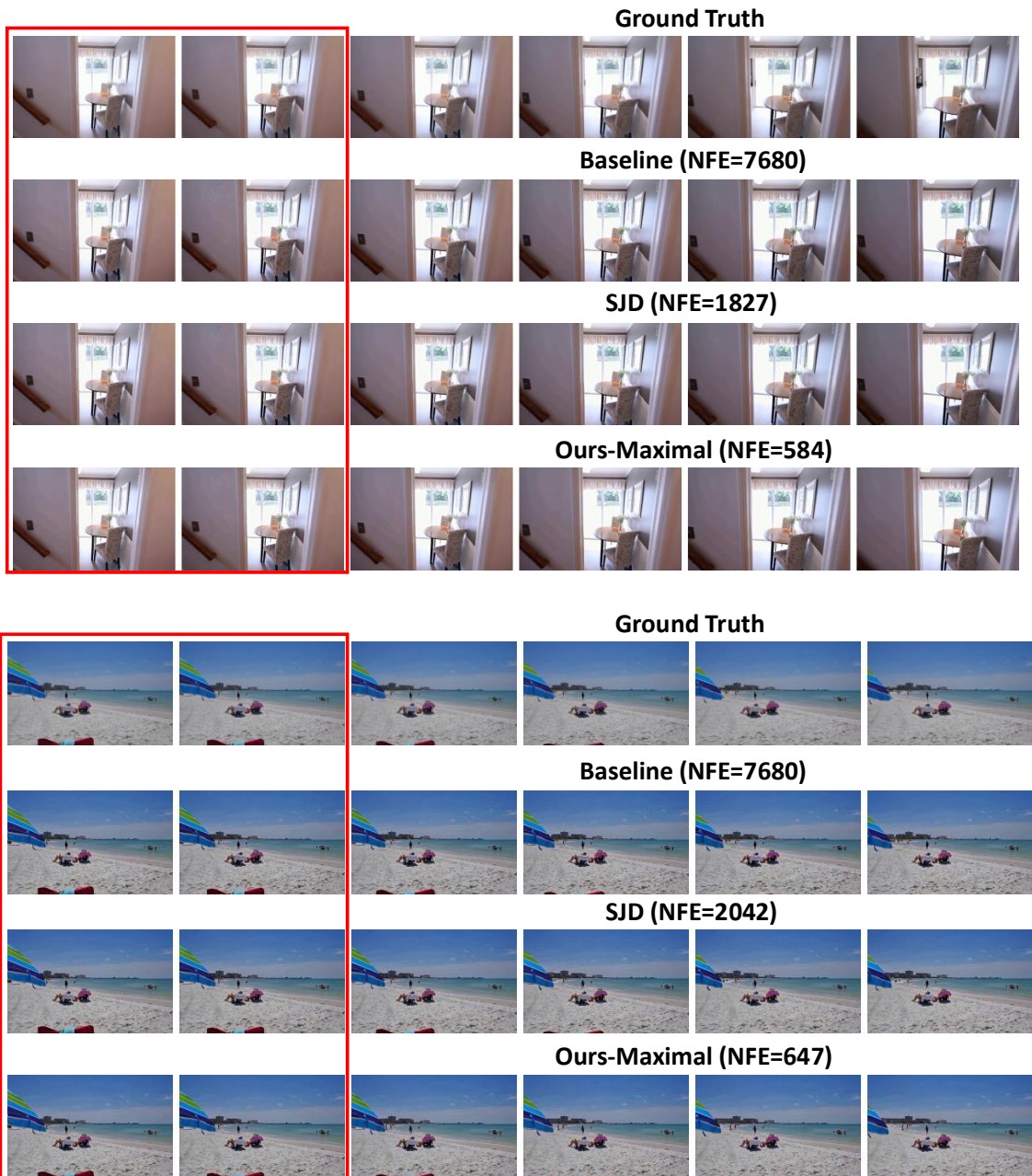

*Figure 11.* Qualitative comparison on Video Generation ( Cosmos-1-ar )

