# OpenReview forum: "Speculative Coupled Decoding for Training-Free Lossless Acceleration of Autoregressive Visual Generation"
_ICML.cc/2026/Conference — ICML 2026 regular_

### Official Review · Reviewer_ybFv · 2026-02-13

**Soundness:** 3
**Presentation:** 3
**Significance:** 3
**Originality:** 3
**Overall Recommendation:** 4
**Confidence:** 2

**Summary:**

This paper studies training-free, lossless acceleration of autoregressive visual token generation via speculative-style iterative decoding. Building on Speculative Jacobi Decoding, the authors identify that independent drafting across iterations causes unstable draft trajectories and high-variance acceptance behavior, limiting speedups. They propose Speculative Coupled Decoding, which replaces independent sampling in the drafting stage with coupled sampling between consecutive iteration distributions, using either maximal coupling or a Gumbel-based coupling. The approach is designed to preserve the correct marginal sampling distribution while increasing the probability that consecutive drafts agree, thereby stabilizing contexts and improving acceptance. Experiments on image and video generation show substantial reductions in required target model evaluations and notable end-to-end speedups, without introducing an extra trained draft model.

**Compliance With Llm Reviewing Policy:**

Affirmed.

**Final Justification:**

The authors have done additional experiments to improve results and provide more concrete evidence of the method working.

**Key Questions For Authors:**

Correctness: Under what exact assumptions does the overall algorithm remain lossless end-to-end? Are there any implementation details (vectorization, caching, truncation, windowing) that could break exactness, and how are they handled?

Coupling choice: In what regimes does maximal coupling outperform Gumbel coupling and vice versa? Can you provide a simple guideline based on measurable properties (e.g., distribution overlap, entropy, iteration stage)?

Sensitivity: How sensitive are the gains to the window length, number of iterations, and any acceptance-related hyperparameters? Is there a robust “default” configuration that works across models?

Scaling: How does performance change as resolution/sequence length increases (more tokens) and as batch size changes? Do the gains persist at higher resolutions or longer videos?

**Limitations:**

yes

**Strengths And Weaknesses:**

Strengths:

Clear and focused idea: a minimal modification to a training-free baseline that targets a well-motivated failure mode (trajectory instability due to independent sampling).

Practical relevance: avoiding an extra draft model is appealing for large-scale visual generators where training or maintaining a second model is costly.

Method is lightweight: coupling mechanisms appear easy to implement and should add limited overhead, making adoption plausible.

Empirical gains are strong on the reported image and video settings, with large reductions in target model evaluations and attractive speedups.

The paper contributes a useful perspective: sampling correlation across iterations can matter as much as distributional closeness for acceptance and convergence in iterative speculative schemes.

Weaknesses:

Limited clarity on the precise “lossless” guarantee under the full pipeline: while coupling preserves marginals by construction, the interaction with the subsequent verification/accept-reject procedure and any implementation shortcuts (e.g., vectorization, windowing) could benefit from a more explicit, end-to-end correctness argument and boundary conditions.

Generality is not fully established: results are shown on a specific set of AR visual models/datasets; it is unclear how robust the gains are across other model families, tokenizers, conditioning types (text-to-image vs. image-to-video vs. video continuation), and decoding hyperparameters.

Ablations and sensitivity: the method introduces choices (coupling type, window length, iteration budgets, thresholds). The paper could more systematically characterize when maximal vs. Gumbel coupling is preferable, and how performance changes with these knobs.

Quality evaluation for video: relying on a single main metric can be insufficient; more complementary perceptual or task-based evaluations and qualitative failure analysis would strengthen the claim that quality is preserved in practice.

Runtime accounting: the speedup story would be stronger with a clearer breakdown of where time is spent (drafting, verification, coupling overhead, memory movement), including hardware details and batch/sequence-length scaling.

---

> ### Author Rebuttal · Authors · 2026-03-31
>
> Thank you for your positive assessment and valuable feedback.
>
> ---
>
> **W1, Q1: Correctness clarity:**
>
> Our exactness guarantee relies on two assumptions: (i) draft token follows the correct marginal, and (ii) uses the same verification rule as in standard SD. Because our coupling does not violate either, the single-token exactness is guaranteed. By induction, the subsequent accept/reject procedure remains lossless, since the same argument applies sequentially over the drafted prefix until the first rejection.
>
> Vectorization and windowing are only code-level implementations and do not change the underlying sampling rule, so they also do not affect exactness. For example, vectorization is just a parallelized for-loop of tokenwise MRS while preserving the same first-rejection behavior as the original. We will clarify it in the final version.
>
> ---
>
> **W2: Generality of the Method:**
>
> We believe our method is evaluated on a fairly broad set of settings, including 2 modalities (image , video), 4 model families/tokenizers (Lumina 1, Lumina 2, Janus-Pro, Cosmos-1-AR), and 2 datasets (MS-COCO, Parti-Prompt), sufficiently showing our method is not restricted to a single setting.
>
> To test conditioning robustness, we newly perform image-to-image and image-to-video experiments in below.
>
> | image2image (Lumina2) | NFE↓| Latency(s) |
> |---|---:|---:|
> | AR |4162| 564.9|
> | SJD(L=64) |1973.5| 165.7|
> | **Ours(GS)** (L=64) |1323.9 | 119.8|
> | **Ours(MC)** (L=64) |1275.9 | 107.1|
>
> | image2video (Cosmos1) | NFE ↓ | Latency(s) |
> |---|---:|---:|
> | AR |7680| 158.0|
> | SJD(L=128) |2383|51.5|
> | **Ours(MC)** (L=128) |810|17.5|
> | **Ours(GS)** (L=128) |792|18.10|
>
> For i2i, we generated 200 images from their depth maps. For i2v, we modified the number of conditioning frames from 9 to 1. As shown, our method still provides strong improvements under these additional conditioning types.
>
> ---
>
> **W3, Q2: Coupling type choice:**
>
> > In what regimes does MC outperform Gumbel? Can you provide guideline?
>
> As a simple guideline, we recommend using *Gumbel* by default and considering *MC* for complex tasks. Please see our answer to Reviewer AEpb Q3.
>
> ---
>
> **W3, Q3: Hyperparameter sensitivity:**
>
> Except for the coupling type, our algorithm does not introduce any additional hyperparameters compared to SJD (# iterations are determined by the decoding process). Moreover, our SCD is not sensitive to the window size, as it achieves monotonic speedup as the window size increases. By default, we recommend using \(L=32\), as this window can be fully parallelized on modern GPUs without noticeable overhead.
>
> Besides, some decoding parameters, such as CFG scale, Top-K, and temperature, can affect performance because they modify target distribution. To check sensitivity, we report the performance of our method under different Top-K settings :
>
> | (NFE ↓) | K=10 |K=500 | K=2000 (Default) |
> |---:|---:|---:|---:|
> |SJD|936.7|1045.2|1035.9|
> |**SCD (Ours)**|596.2|565.0|567.7|
>
> As shown, SCD remains robust performance under different Top-K. Although speedup slightly diminishes for small K, such settings are uncommon in visual AR because they significantly degrade generation quality.
>
> ---
>
> **W4: More quality metrics for video:**
>
> Although we have theoretically shown that our method is lossless, we agree that additional empirical evidence can strengthen our claim. We evaluate 4 additional metrics: DreamSim [1] as a high-level alignment, and PSNR, SSIM, and LPIPS as low-level metrics.
>
> |Method|DreamSim ↑ | PSNR ↑ | SSIM ↑ | LPIPS ↓|
> |---|---:|---:|---:|---:|
> |AR | 0.936 | 22.685 | 0.744 | 0.208 |
> |SJD (L=128) | 0.938 | 22.804 | 0.745 | 0.205 |
> |**Ours_MC** (L=128) | 0.937 | 22.796 | 0.746 | 0.205 |
> |**Ours_GS** (L=128) | 0.936 | 22.749 | 0.743 | 0.208 |
>
> As shown, our method achieves scores nearly identical to those of AR, empirically supporting our exactness claim.
>
> ---
>
>
> **W5: Runtime Breakdown:**
>
> Please see Sec. 4.3 and Tab. 2 in main paper.  We present hardware details (RTX-3090) and a latency breakdown of each component in a single NFE step.
>
> ---
>
>
> **W5,Q4: Scaling with sequence length:**
>
> To examine the scaling effect, we evaluate with Lumina-mGPT at different generation resolutions.
>
> | (NFE ↓)   | AR | SJD (L=16) | **Ours (L=16)** | SJD (L=32) | **Ours (L=32)** | SJD (L=64) | **Ours (L=64)** |
> |---|---|---|---|---|---|---|---|
> | 512×512 |  1131 | 421.9 | **357.2** | 443.5 | **278.5** | 443.7 | **211.5** |
> | 768×768 |  2390 | 1058.6 |**814.5**| 1031.2 | **666.0** | 1035.9 | **567.7** |
> | 1024×1024 |  4237 | 1765.9 |**1284.1**| 1771.8 | **988.7** | 1735.5 | **785.3** |
>
> As shown, the gains of our method persist as the sequence length increases.
>
> **Video length scaling :** As Cosmos-1 supports only a single duration, we are unable to report results for longer video lengths (we report the maximum duration in the paper). We kindly ask for your understanding on this point.
>
> ---
>
> [1] DreamSim: Learning New Dimensions of Human Visual Similarity using Synthetic Data, NeurIPS23

---

> > ### Author Rebuttal · Reviewer_ybFv · 2026-04-01
> >
> > My concerns have been adequately addressed.

---

> > > ### Author Response · Authors · 2026-04-06
> > >
> > > Thank you for your positive evaluation and engagement in the discussion. We are very pleased that our responses have fully addressed your concerns. If you feel it is appropriate, we would greatly appreciate your consideration of a score update.
> > >
> > >
> > >
> > > Thank you for your time and consideration.

---

### Official Review · Reviewer_abkf · 2026-03-09

**Soundness:** 4
**Presentation:** 3
**Significance:** 4
**Originality:** 3
**Overall Recommendation:** 4
**Confidence:** 3

**Summary:**

The paper finds that the performance of the Speculative Jacobi Decoding (SJD) is significantly limited by instability in its draft token sampling.

Than the paper proposes a information-theory-inspired idea, which couples the draft sampling process between consecutive Jacobi iterations, increasing the probability of sampling identical tokens to promote stability.

The proposed method requires only a single-line modification to standard SJD, making it extremely simple to implement.

The proposed method significantly enhances the acceptance rate of SJD, enabling a remarkable speedup of ∼3.8× for AR image generation and ∼10× for video generation compared to standard AR decoding.

**Compliance With Llm Reviewing Policy:**

Affirmed.

**Final Justification:**

All my concerns are solved.

**Key Questions For Authors:**

How can the proposed method improve more advanced Speculative Decoding methods like GSD?

**Limitations:**

yes

**Strengths And Weaknesses:**

Strength
The observation on the limitation of Speculative Jacobi Decoding (SJD) is insightful, and is analyzed by bounding the collision probability.

Based on above obsevation, the proposed coupling method formalized by information theory, and is very easy to be implemented.

Only one line of code is changed on standard Speculative Jacobi Decoding.

The acceleration over standard SJD is significant with different models including Lumina-mGPT, Janus-Pro (7B), Cosmos1-AR-4B for both image and video generation.




Weakness
One limitation would be how the proposed method can improve more advanced Speculative Decoding methods like GSD. Currently, the experiments are all conducted on standard speculative jacobi decoding.

---

> ### Author Rebuttal · Authors · 2026-03-31
>
> We appreciate your positive assessment. We have provided detailed responses to your questions below.
>
> ---
>
> **W1, Q1: Integration with `Grouped Speculative Decoding` :**
>
> This is a very interesting point. Because GSD is applied at the *verification* stage of speculative decoding, while our SCD is applied at the *drafting* stage, the two methods are complementary and can be combined synergistically. We show the results of combining GSD and SCD in the tables below.
>
> ---
>
> *Lumina-mGPT (Image Generation)*
>
> | Method | FID ↓ | NFE ↓ |
> |---|---:|---:|
> | AR | 30.79 | 2390 |
> | SJD (L=32) | 30.78 | 1031.2 |
> | SJD (L=32) + GSD (G=3) | 31.50 | 925.9 |
> | SJD (L=32) + GSD (G=10) | 33.21 | 701.4 |
> | **SCD (L=32)** | 30.78 | **657.2** |
> | **SCD (L=32) + GSD (G=3)** | 30.99 | **416.65** |
> | **SCD (L=32) + GSD (G=10)** | 32.91 | **298.2** |
>
> ---
>
> *Cosmos1-AR (Video Generation)*
>
> | Method | FVD ↓ | NFE ↓ |
> |---|---:|---:|
> | AR | 170.3 | 7680 |
> | SJD (L=128) | 169.8 | 1786 |
> | SJD (L=128) + GSD (G=3) | 165.5 | 1030 |
> | SJD (L=128) + GSD (G=30) | 171.0 | 926 |
> | **SCD (L=128)** | 168.2 | **565** |
> | **SCD (L=128) + GSD (G=3)** | 166.1 | **455** |
> | **SCD (L=128) + GSD (G=30)** | 172.7 | **444** |
>
> ---
>
> As shown in the tables, combining GSD and SCD yields substantial additional acceleration at the cost of a small drop in generation quality. We believe that combining our method with such lossy acceleration schemes is a highly promising direction for future work, and we plan to include these results in the final version.

---

> > ### Author Rebuttal · Reviewer_abkf · 2026-04-02
> >
> > All my concerns are solved.

---

> > > ### Author Response · Authors · 2026-04-06
> > >
> > > Thank you for your positive evaluation and suggesting insightful feedback in the discussion. We are very pleased that our responses have fully addressed your concerns. If you feel it is appropriate, we would greatly appreciate your consideration of a score update.
> > >
> > >
> > > Thank you for your time and consideration.

---

### Official Review · Reviewer_54io · 2026-03-13

**Soundness:** 3
**Presentation:** 3
**Significance:** 3
**Originality:** 3
**Overall Recommendation:** 4
**Confidence:** 4

**Summary:**

A pertinent problem studied by this manuscript is the slow token-by-token inference of autoregressive visual generators, where thousands of steps may be needed for one image or video. Overall, this paper's principal aspect is a training-free modification to Speculative Jacobi Decoding (SJD) that aims to keep exact autoregressive sampling while improving speed. The paper argues that SJD underperforms because independently sampled draft tokens create context instability across Jacobi iterations, which lowers acceptance rates. It proposes Speculative Coupled Decoding (SCD), replacing independent draft sampling with coupling between consecutive iteration distributions; the two instantiations are maximal coupling via modified rejection sampling and Gumbel-sharing coupling (Sec. 4, Algs. 3-4). The paper claims losslessness from marginal-preservation arguments (Theorems 4.2, 4.4, 4.5) and low overhead. Empirically, it reports up to 4.2x image-generation speedup on Lumina-mGPT and large video NFE reduction on Cosmos-1-AR (7680 to 564.4 NFEs, with latency 157.25s to 13.60s), while keeping quality metrics close to AR in Tables 1, 3, and 4.

**Compliance With Llm Reviewing Policy:**

Affirmed.

**Key Questions For Authors:**

Please see the Weaknesses. I encourage the authors to address the concerns outlined above in the rebuttal. If these issues are satisfactorily resolved, I would be open to revising my assessment and increasing my final score.

**Limitations:**

The authors  should discuss the limitations and potential negative societal impact of their work

**Strengths And Weaknesses:**

[S1] The paper provides a concrete diagnosis of SJD’s bottleneck rather than only presenting a new sampler. The link between acceptance rate, context similarity, and token collision probability is developed analytically (Proposition 2.1, Proposition 3.3, Eqs. 1-2).

[S2] The methodological change is simple and tightly integrated with existing speculative decoding machinery. In Sec. 4, the core change is to the drafting step (Alg. 3), maximal coupling reuses the same MRS primitive already used for verification (Sec. 4.1, pp. 5-6), and the paper gives explicit correctness statements for both maximal and Gumbel variants (Theorems 4.2, 4.4, 4.5).

[S3] Good experimental results. On Lumina-mGPT, the best setting reduces latency from 102.03s to 24.24s and NFEs from 2390 to 568.0, while FID/IS/CLIP remain very close to vanilla AR; the shown lossy GSD settings are slower or similar-speed but noticeably worse on FID/IS (Table 1, p. 7). The evaluation is not confined to one image model. The paper also reports gains on Janus-Pro, where NFEs drop from 576 to 154.42 and latency from 13.218s to 5.388s (Table 3, p. 7), and on Cosmos-1-AR video generation, where NFEs drop from 7680 to 564.4 and latency from 157.25s to 13.60s in the best reported setting (Table 4, p. 8).

[W1] Part of the causal argument is left informal. Observation 3.1 relies on a “mild Lipschitzness assumption” to argue that more similar contexts induce more similar output distributions (p. 3), but that assumption is neither formalized nor tested for the image/video AR models used here.

[W2] Some reported quality numbers are slightly worse than vanilla AR, e.g., in Table 1 the Lumina-mGPT FID is 30.79 for AR versus 30.90 for πGS at L=64, and in Table 3 the Janus-Pro FID is 37.96 for AR versus 38.34 for πMC at L=32 (p. 7).

---

> ### Author Rebuttal · Authors · 2026-03-31
>
> We truly appreciate your positive assessment and insightful feedback. We have provided detailed responses to your questions below.
>
> ---
>
>
> **W1: Lipschitzness assumption:**
>
> Thank you for the insightful comment. We would like to argue that *Observation 3.1* is empirically tested through *Fig.3*.  In particular, *Fig.3* was obtained on Lumina-mGPT with window size \(L=64\), where we plot the mean token difference between consecutive drafts against the resulting generation NFE over 300 independent samples. As *Eq.1* shows that the acceptance rate is governed by the output discrepancy between consecutive iterations, this empirical observation—that smaller input differences are associated with lower NFEs ( lower output discrepancy )—supports our Lipschitzness assumption in practice.
>
> We consider this empirical validation to be more meaningful than explicitly defining a Lipschitz constant for a large visual transformer, whose input is discrete while the output is a high-dimensional continuous distribution. Moreover, this observation is used only to motivate why stabilizing the draft context can improve acceptance, and does not affect our theorems on exactness. We hope this explanation addresses your concern.
>
> ---
>
> **W2: FID difference:**
>
> The FID score can have slight fluctuations due to finite-sample estimation noise, and these small differences are within the error range. To verify this, we additionally evaluated the method with 5 new random seeds and report the mean and standard deviation of the FID score.
>
> *Lumina-mGPT*
>
> | (FID ↓) | Seed 15 | Seed 20 | Seed 25 | Seed 35 | Seed 45 | ***Mean*** | ***Std.*** |
> |---|---:|---:|---:|---:|---:|---:|---:|
> | AR | 30.715 | 30.493 | 31.211 | 30.895 | 30.974 | **30.857** | **0.270** |
> | **SCD(GS)** (L=64) | 30.836 | 31.305 | 30.929 | 30.406 | 30.998 | **30.894** | **0.324** |
>
> *Janus-Pro*
>
> | (FID ↓) | Seed 15 | Seed 20 | Seed 25 | Seed 35 | Seed 45 | ***Mean*** | ***Std.*** |
> |---|---:|---:|---:|---:|---:|---:|---:|
> | AR | 37.978 | 38.240 | 38.219 | 37.449 | 38.495 | **38.076** | **0.628** |
> | **SCD(MC)** (L=32) | 37.884 | 37.906 | 38.438 | 38.160 | 37.938 | **38.065** | **0.485** |
>
> As shown in the table, the FID scores of vanilla AR and our method remain within the error range. We will update the main tables with error statistics in the final version. Thanks for pointing this important point.
>
> ---

---

### Official Review · Reviewer_AEpb · 2026-03-13

**Soundness:** 3
**Presentation:** 3
**Significance:** 3
**Originality:** 3
**Overall Recommendation:** 5
**Confidence:** 2

**Summary:**

This paper proposes Speculative Coupled Decoding (SCD) to speed up autoregressive image and video generation. It improves Speculative Jacobi Decoding (SJD) by using coupled sampling so that consecutive drafts sample the same tokens more often when their distributions are similar. This stabilizes the draft sequence, increases the acceptance rate during verification, and reduces the number of model evaluations. The method requires no extra training and achieves up to 4.2× speedup for images and 13.6× for videos without hurting generation quality.

**Compliance With Llm Reviewing Policy:**

Affirmed.

**Final Justification:**

My concern has been addressed, so i will maintain my score.

**Key Questions For Authors:**

1. The evaluation primarily focuses on visual autoregressive models (image and video generation). How does the proposed coupling strategy perform for LLMs or other modalities?

2. The effectiveness of the proposed coupling strategy relies on consecutive draft distributions being similar. However, in image and video generation, some regions may contain high-frequency details (e.g., textures, edges, or rapid motion in videos), where token distributions may change more significantly across iterations. How does the method perform in such high-frequency regions, and does the coupling strategy still provide stable drafts in these cases?
3.  The paper introduces two coupling strategies (maximal coupling and Gumbel coupling). Could the authors provide deeper insights into when one coupling method should be preferred over the other in practice?

**Limitations:**

yes

**Strengths And Weaknesses:**

### Strengths
- Introduces a simple and elegant improvement to Speculative Jacobi Decoding by using coupled sampling to stabilize draft tokens.
- Training-free and lossless, preserving the exact output distribution of autoregressive decoding without requiring a separate draft model.
- Very lightweight to implement with negligible computational overhead.
- Demonstrates substantial empirical speedups.
- Provides clear theoretical analysis explaining why independent sampling in SJD causes instability and how coupling improves acceptance rates.

### Weaknesses

- The method still inherits some limitations of SJD, such as sensitivity to window size and iterative refinement behavior.
- Evaluation focuses mainly on visual autoregressive models, and it is unclear how well the approach generalizes to other domains such as text LLM decoding.
- Effectiveness relies on consecutive draft distributions being similar, which may not always hold for more complex or highly uncertain generation tasks.

---

> ### Author Rebuttal · Authors · 2026-03-31
>
> We sincerely appreciate your positive assessment of our contribution. Below, we provide detailed responses to your questions.
>
> ---
>
> **W1 : Inherit SJD's weakness:**
>
> Thank you for pointing this. We would like to clarify that our SCD is actually designed to ***resolve*** these limitations of SJD.
>
> - **Iterative refinement** : Iterative refinement is a desirable property in fixed-point systems such as Jacobi decoding, as it enables rapid convergence to the solution. However, the high sampling variance of SJD causes the draft context to effectively "reset" at almost every iteration, preventing meaningful refinement. In contrast, our coupling reduces this variance, allowing the model to refine its context and thereby achieve higher acceptance rates.
> - **Window size sensitivity** : It is true that standard SJD suffers from window-size sensitivity, as larger window sizes do not reliably translate into better speedups. In such cases, users must carefully tune the window size to balance hardware overhead and acceptance rate. In contrast, our SCD effectively leverages larger window sizes to deliver monotonically increasing acceleration through its refinement behavior. This allows users to simply choose the largest window size their hardware memory permits, removing the burden of window-size tuning.
>
> ---
> **W2, Q1 : Applicability to other modalities:**
>
> We expect our method to apply broadly to other modalities that use Vector Quantization (VQ) for tokenization. Since VQ discretizes continuous data, different codebook entries often share overlapping semantics, leading to high-entropy next-token distributions. Our SCD is particularly effective in this regime, where high entropy causes draft token instability through low collision probabilities.
>
> To demonstrate this, we conducted an additional experiment on a Text-To-Speech (TTS) task by using Audio LLM, Cosyvoice2 [1] on LibriTTS dataset.
>
> | Method | SpeedUp ↑ | WER ↓ |
> |---|---:|---:|
> | AR |  1.00x | 3.49% |
> | **SCD(L=8)** | 1.32x | 3.30% |
> | **SCD(L=16)** | 1.33x| 3.68% |
> | **SCD(L=32)** | 1.39x| 3.30% |
>
> The results show that our SCD consistently accelerates the decoding process in audio AR. We think applying our method to other VQ-based AR models (e.g., 3D, Motion) is a highly promising  future work.
>
> Besides, we do not observe meaningful speedups in the text AR. This is because text models usually have a one-hot-like next-token distribution due to the structural nature of text data (Zipf's law) and low-temperature decoding. In these cases, the room for randomness—from SD acceptance to draft token coupling—disappears, converting the entire decoding process into a static argmax decoding process.
>
> ---
>
> **W3, Q2 : Coupling effect on high-frequency pattern:**
>
> Thank you for the insightful question. To examine whether SCD remains effective on high-frequency patterns, we conducted the following additional experiments on both image and video generation:
>
> - **Image** : Using Lumina-mGPT, we compared 25 image prompts consisting of simple patterns (e.g., "simple background...") against 25 prompts containing high-frequency patterns (e.g., "curly hair ...") from Parti-Prompts.
> - **Video** : Using Cosmos-1 on the RealEstate10K dataset, we compared the top 25% vs. the bottom 25% of videos based on the VBench [2] dynamic degree metric.
>
> | Modality | Method | Type | NFE ↓ |
> |---|---:|---:|---:|
> | Image | AR | - | 2304 |
> |  | SJD (L=32) | Simple | 1048.2 |
> |  | **Ours (L=32)** | Simple | **545.1** |
> |  | SJD (L=32) | Dynamic | 1088.15 |
> |  | **Ours (L=32)** | Dynamic | **578.2** |
> | Video | AR | - | 7680 |
> |  | SJD (L=128) | Static | 1694.0 |
> |  | **Ours (L=128)** | Static | **539.9** |
> |  | SJD (L=128) | Rapid | 1856.6 |
> |  | **Ours (L=128)** | Rapid | **596.2** |
>
> As shown in the table, the dynamic/rapid cases do show slightly higher NFEs but the gap relative to the corresponding SJD baseline remains highly consistent, and is even slightly better in some cases. In practice, we observed that the acceptance rate does not vary drastically depending on detailed characteristics (e.g. background , hair ...).
>
> ---
>
> **Q3: Coupling Type Choice:**
>
> As a simple guideline, we recommend using GS by default and considering MC for complex generation tasks. As discussed in *Sec.5.1*, GS promotes long-range stability, which leads to clearly better NFEs in tasks where draft prediction is easy, such as video AR (*Table 4*). Meanwhile, GS maintains comparable NFEs even in tasks where maximal coupling (MC) performs slightly better (*Table 1*).
>
> However, please note that our main purpose in presenting two distinct coupling methods was to show that different implementations of our theoretical foundation yield consistent performance improvements, thereby providing robust empirical support for our main claim.
>
> ---
>
> [1] Cosyvoice 2: Scalable streaming speech synthesis with large language models." Arxiv 24.
> [2] Vbench: Comprehensive benchmark suite for video generative models, CVPR 2024.

---

> > ### Author Rebuttal · Reviewer_AEpb · 2026-04-02
> >
> > Thanks for the response. My concern has been fully addressed.

---

> > > ### Author Response · Authors · 2026-04-06
> > >
> > > Thank you for your positive evaluation and engagement in the discussion. We are very pleased that our responses have fully addressed your concerns. We sincerely appreciate your support and thoughtful consideration.

---

### Decision · Program_Chairs · 2026-04-30

**Decision:**

Accept (regular)

**Comment:**

This work presents Speculative Coupled Decoding (SCD), a training-free method improving over the prior Speculative Jacobi Decoding (SJD) for autoregressive visual generation. The improvement ingredients include replacing independent draft sampling with coupled sampling (maximal or Gumbel). The resulting lightweight method requires only a single-line modification and achieves significant speedups.

While all reviewers appreciate the method's elegant simplicity and strong empirical results, they also raise several concerns (e.g., its generalizability to other modalities and sensitivity to hyperparameters). However, the provided rebuttal effectively assuaged the reviewer concerns. After checking the submission, reviewer comments, and author rebuttals, the AC appreciates the insightful diagnosis of SJD's bottlenecks and the highly practical nature of the proposed solution. As a result, the AC recommends acceptance for the submission and encourages the authors to incorporate the rebuttal experiments into their final work.